# Offline Multi-agent Continual Cooperation via Skill Partition and Reuse

**Yuchen Xiao** [1 2]   **Lei Yuan** [1 2 3]   **Ruiqi Xue** [1 2]   **Tieyue Yin** [4]   **Yang Yu** [1 2 3]

## Abstract

Extracting skills from multi-agent offline dataset improves learning efficiency via sharing task-invariant coordination skills among tasks. In settings where tasks occur sequentially and the space of skills grows exponentially, existing approaches that rely on heuristically designed and fixed-sized skill libraries struggle to resolve the problem of distributional shift and interference, facing catastrophic forgetting and plasticity loss. To address this problem and endow agents with the ability to continually discover and reuse coordination skills in open-environment, we propose COMAD, a principled framework for **C**ontinual **O**ffline **M**ulti-**a**gent Skill **D**iscovery via Skill Partition and Reuse. We first discover skills from mixed multi-agent behavior data with an auto-encoder to transform coordination knowledge into reusable coordination skills. Then we construct a skill-augmented policy learning objective with multi-head architectures, explicitly guiding the advantage function with reusable skills identified via a density-based reusability estimator. Theoretical analysis shows our method approximates the optimum of a continual skill discovery problem. Empirical results across diverse MARL benchmarks show that COMAD continually expands its skill library to mitigate interference, achieving superior forward and backward transfer for task streams compared to multiple baselines. Code is available at https://github.com/yuchen2003/comad-icml26.

---

[1]National Key Laboratory for Novel Software Technology, Nanjing University [2]School of Artificial Intelligence, Nanjing University [3]Polixir Technologies. [4]Kuang Yaming Honors School, Nanjing University. Correspondence to: Lei Yuan <yuanl@lamda.nju.edu.cn>.

*Proceedings of the 43$^{rd}$ International Conference on Machine Learning*, Seoul, South Korea. PMLR 306, 2026. Copyright 2026 by the author(s).

## 1. Introduction

Cooperative Multi-agent Reinforcement Learning (MARL) has achieved remarkable success in solving complex cooperative tasks, such as transportation scheduling, robotics, and games (Albrecht et al., 2024). Yet, the prohibitively high sample complexity of online interactions impedes its deployment in real-world applications. Consequently, offline Reinforcement Learning (RL) (Levine et al., 2020) and offline MARL (Prudencio et al., 2023) have emerged as promising paradigms that extract optimal policies directly from pre-collected datasets. Unlike the online setting, where agents can improve their policies in a trial-and-error manner, offline agents face the challenge of distribution shift between the learned policy and the behavior policy. This challenge is further amplified in MARL due to the curse of dimensionality: as the joint state-action space grows exponentially with the number of agents, the extrapolation error caused by distribution shift becomes increasingly severe.

By treating action chunks as abstractions across the temporal dimension, skill discovery has demonstrated significant improvements in sample efficiency within complex settings (Pateria et al., 2021; Mohan et al., 2024). Typical methods primarily discover skills without a reward function in an unsupervised manner, covering the action space to facilitate downstream policy learning and improve generalization (Eysenbach et al., 2018). Some works focus on skill discovery from offline data to enhance learning efficiency by learning temporal action embeddings (Ajay et al., 2021; Kim et al., 2023); these methods heuristically design a fixed-size embedding space to span the skill space, outperforming naive RL in sample efficiency. Unlike single-agent scenarios, offline skill discovery in multi-agent systems is more complex due to agent interactions. Previous works mainly discover skills at the agent level by learning temporal abstractions—similar to the single-agent setting (Chen et al., 2024b; Meng et al., 2023)—or at the team level to learn spatial coordination skills (Zhang et al., 2023; Liu et al., 2025), showing improvements in both learning efficiency and generalization.

However, the aforementioned methods implicitly assume that all necessary coordination skills can be captured within a pre-defined skill library, limiting their applicability in open-ended environments where new tasks arrive sequen-

tially (Yuan et al., 2023). Concretely, when agents learn in an open-ended setting, the required skills may shift according to task characteristics. Agents may lose their plasticity to accommodate emerging skills, or suffer from severe interference when directly fine-tuning skill representations, inevitably leading to catastrophic forgetting of previously acquired skills. To address this, works such as Lotus (Wan et al., 2024) construct an ever-growing skill library from a sequence of new tasks using limited human demonstrations, while TAIL (Liu et al., 2024) designs Task-specific Adapters for Imitation Learning, demonstrating high adaptability. Although these methods are effective to some extent, they rely heavily on pre-trained models, which is often ill-suited for the MARL, where tasks vary not only in the underlying MDP but also in their high-level semantics, leading to distinct equilibrium solutions (Zhang et al., 2021).

To endow agents with the ability to continuously learn coordination skills from offline data, agents should identify the tasks correctly and retrieve task-invariant coordination patterns hidden within diverse tasks. We first formalize the problem as learning from an infinite stream of multi-agent tasks, modeled as a series of decentralized partially observable Markov decision processes (Dec-POMDPs). We then propose a principled framework for Continual Offline Multi-agent Skill Discovery via skill partition and reuse (COMAD). COMAD partitions different skills using multi-head architectures and selectively reuses them via reusability estimation. To discover skills from task-specific behavior data, we utilize a variational auto-encoder (VAE) (Kingma & Welling, 2013) to encode global state-action pairs into skill representations and decode them via an action decoder, while a local encoder infers skills using only local information. Crucially, to handle the expanding skill space, we design an actor equipped with multi-head modules and a reusability estimator. These modules allow us to estimate the reusability of previously acquired skills for new tasks and leverage guidance from these skills for effective transfer and deployment. Our theoretical analysis reveals that skills can be optimally transferred via reusability estimation and skill guidance. Extensive experiments on distinct multi-agent environments demonstrate the superior performance compared to different baselines.

## 2. Related Work

**Multi-agent Reinforcement Learning**  utilizes RL to address multi-agent problems. Unlike single-agent settings, MARL faces the curse of dimensionality in the joint state-action space, which stems from the growing number of agents. To overcome this challenge, typical works utilize value decomposition (Sunehag et al., 2017; Rashid et al., 2018) or policy decomposition (Wang et al., 2021) to transform the complex high-dimensional joint space into

tractable low-dimensional representations, demonstrating high learning efficiency in fields such as autonomous driving, financial trading, and embodied intelligence (Feng et al., 2025). Nevertheless, the practical utility of online MARL methods is limited by the requirement for costly online interactions. Consequently, offline MARL has emerged to enhance data efficiency by learning directly from historical data. Typical approaches focus on frameworks that integrate multi-agent decomposition methods with implicit regularization. For instance, Implicit Constraint Q-learning (ICQ) (Yang et al., 2021) proposes to implicitly constrain local policy learning via a constrained Q-learning objective. Furthermore, OMIGA (Wang et al., 2023) applies a behavior-regularized framework to implicitly constrain the value function, deriving an elegant objective that bypasses the computationally intractable partition function.

**Continual Reinforcement Learning**  addresses the challenge of learning from a sequential stream of tasks in open-ended environments (Zhou, 2022), where environmental dynamics and objectives may evolve dynamically. In such settings, agents must maintain a critical balance between stability (preserving memory of previous tasks) and plasticity (the ability to learn new tasks) (Pan et al., 2025). Typical approaches include parameter regularization (Kirkpatrick et al., 2017), experience replay (Chen et al., 2024a), parameter isolation (Kessler et al., 2022), and gradient projection (Saha & Roy, 2023), demonstrating promise in enabling effective decision-making within open-ended environments. While the majority of existing methods focus on single-agent settings, several recent works have directed attention toward continual MARL. For instance, MACPro (Yuan et al., 2024) proposes to identify task contexts and dynamically expand the network architecture for adaptation in an online way. Similarly, RPG (Yao et al., 2025) extracts task-agnostic relational patterns and subsequently generates task-specific decision-makers via a conditional hypernetwork to guide adaptation, showing the continual learning ability in MARL.

**Skill Discovery**  learns action chunks via temporal abstraction, typically adopting hierarchical structures to guide the agent's decision-making (Pateria et al., 2021; Mohan et al., 2024). Foundational methods generally discover skills via unsupervised learning objectives, utilizing them to promote exploration (Eysenbach et al., 2018) or facilitate downstream task adaptation (Wan et al., 2024). Recent advancements have extended this paradigm to offline data; for instance, HILP (Park et al., 2024) learns Hilbert representations to enable zero-shot adaptation. Furthermore, skill discovery in MARL focuses on decomposing the complex joint action space into composable subspaces to facilitate cross-agent and cross-task transfer. Specifically, ODIS (Zhang et al., 2023) first discovers skills via a Variational Auto-Encoder (VAE) and subsequently learns a coordination pol-

icy based on these latent skills. Similarly, HiSSD (Liu et al., 2025) learns both common skills and task-specific skills to enable fine-grained multi-agent cooperation. However, current approaches often assume a static and fixed skill library, thereby limiting their applicability in open-ended environments, where new task scenarios and novel skills may continuously emerge (Yuan et al., 2023).

## 3. Problem Formulation

This paper considers the offline multi-agent skill discovery problem, formulated as a decentralized partially observable Markov decision process (Dec-POMDPs) (Oliehoek et al., 2016):

$$\mathcal{M} = \langle N, S, A, \Omega, O, R, P, \gamma \rangle,$$

which is composed of agent team $N$, state space $S$, action space $A$, observation space $\Omega$ with observation function $O : S \times N \to \Omega$, reward function $R : S \times A \to \mathbb{R}$ and transition function $P : S \times A \to \Delta(S)$ that maps pairs of state and action to the distributions of next states. $\gamma \in [0, 1)$ is the discount factor. The goal is to find a policy $\pi(\boldsymbol{a}|s)$ to maximize the discounted return $J(\pi) = \mathbb{E}_{P,\pi}[\sum_t \gamma^t R(s_t, \boldsymbol{a}_t)]$. For the optimization process, we use the implicit Q-Learning pipeline for its excellent learning efficiency in single agent (Kostrikov et al., 2022) and multi-agent setting (Wang et al., 2023) as:

$$Q^{tot}(\boldsymbol{o}, \boldsymbol{a}) = \sum_i w^i(\boldsymbol{o}) Q^i(o^i, a^i) + b(\boldsymbol{o}),$$

$$V^{tot}(\boldsymbol{o}) = \sum_i w^i(\boldsymbol{o}) V^i(o^i) + b(\boldsymbol{o}), \qquad (1)$$

$$w^i \geq 0, i = 1, \cdots, n.$$

For MARL, the value functions are learned via the following behavior constrained objective:

$$\min_{\substack{Q^i, w^i, b \\ i=1,\cdots,n}} L_Q := \qquad\qquad (2)$$

$$\mathbb{E}_{(\boldsymbol{o}, \boldsymbol{a}, \boldsymbol{o}') \sim D} \left[ (r(\boldsymbol{o}, \boldsymbol{a}) + \gamma V^{tot}(\boldsymbol{o}') - Q^{tot}(\boldsymbol{o}, \boldsymbol{a}))^2 \right],$$

$$\min_{V^i} L_V := \mathbb{E}_{a_i \sim \mu_i} \left[ \frac{w^i(\boldsymbol{o})}{\alpha} V^i(o^i) \qquad (3) \right.$$

$$\left. + \exp\left( \frac{w^i(\boldsymbol{o})}{\alpha} (Q^i(o^i, a^i) - V^i(o^i)) \right) \right],$$

where $\alpha$ is a hyperparameter that controls the strength of behavior regularization.

In continual offline MARL setting, we should extract skills from datasets $D_1, \cdots, D_m, \cdots$ of an infinite stream of multi-agent tasks described by a series of Dec-POMDPs:

$$\mathcal{M}_1, \mathcal{M}_2, \cdots, \mathcal{M}_m, \cdots$$

Furthermore, we adopt the task-bounded setting (Pan et al., 2025), where we assume that the skill $z_m^i$ of agent $i$ in task $m$ is a discrete variable from a finite set $\mathcal{Z}_m$, which encodes different types of individual policies $\pi_m^i(a^i|o^i, z_m^i)$. Skill set $\mathcal{Z}_m$ varies among tasks as new skills emerge and old skills become inefficient due to the shift of action distribution and state distribution, as shown in Figure 1(a). Ideally, to handle the shift, the agent must construct an expandable skill library to flexibly learn new skills while covering and reusing all performant skills for all seen tasks. That is, to maximize the performance $J_m(\pi_m)$ on current task $m$ while maintaining the performance on old tasks above certain thresholds: $J_k(\pi_m) \geq L_k$, where $J_k(\pi_m)$ is the discounted return of $\pi_m$ on task $k$:

$$J_k(\pi_m) = \mathbb{E}_{P_k, \pi_m} \left[ \sum_{t=1}^{\infty} \gamma^t R_k(s_t, \boldsymbol{a}_t) \right], k = 1, \cdots, m.$$

## 4. Method

In this section, we describe our proposed method COMAD in detail. The training cycle proceeds in two stages for each incoming task: the first stage discovers task-specific coordination skills from the offline dataset with an auto-encoder (Section 4.1); and the second stage stores the extracted skills in multi-head models, then selectively reuses them via reusability estimation and a skill-augmented objective (Section 4.2). We further build a continual learning pipeline that integrates the discover-then-transfer process into an efficient algorithm (Section 4.3). Finally, we analyze the effectiveness of this algorithm (Section 4.4).

### 4.1. Discovering Skills from Offline Dataset

A major challenge of learning from multiple multi-agent tasks is that the agent team varies among different tasks, resulting in non-fixed input and output length. To construct a universal continual policy that can adapt to varied tasks, we adopt the widely used population-invariant networks (Hu et al., 2021; Zhang et al., 2023) based on the transformer architecture (Vaswani et al., 2017):

$$Q = \mathrm{MLP}_q([o^{i,\mathrm{env}}, o^{i,1}, \ldots, o^{i,n}])$$
$$K = \mathrm{MLP}_k([o^{i,\mathrm{env}}, o^{i,1}, \ldots, o^{i,n}])$$
$$V = \mathrm{MLP}_v([o^{i,\mathrm{env}}, o^{i,1}, \ldots, o^{i,n}]) \qquad (4)$$
$$[e^{i,\mathrm{env}}, e^{i,1}, \ldots, e^{i,n}] = \mathrm{softmax}\left( \frac{QK^\top}{\sqrt{d_k}} \right) V,$$

where $d_k = \dim(K)$ is the dimension of $K$. Note that the input can also be states $s = [s^{\mathrm{env}}, s^1, \ldots, s^n]$. Via this architecture, the varying-length inputs can be processed into equal-length representations $[e^{i,\mathrm{env}}, e^{i,1}, \ldots, e^{i,n}]$ that can be used to construct value functions and policies.

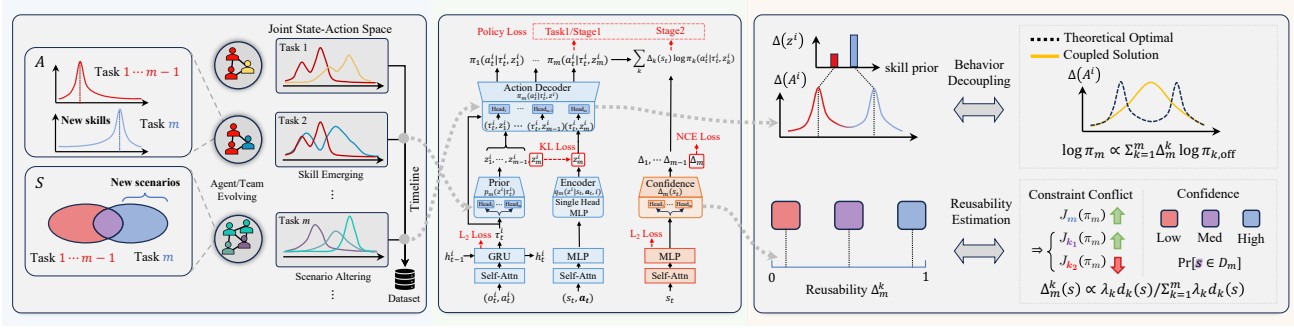

| (a) Multi-agent task stream | (b) Actor Architecture | (c) Three key principles |

*Figure 1.* An illustration of our method COMAD. (a) Given a multi-agent task stream, the continual agents must discover skills from offline data of different tasks sequentially, where two types of distribution shift occur. (b) COMAD actor is equipped with multi-headed architectures to overcome the challenge of emerging skills by encoding and decoding skills from different scenarios, meanwhile estimating their reusability for efficient transfer. (c) The key principles of COMAD for continual skill discovery include decoupling behaviors from mixed datasets and selectively reuse them based on reusability.

To effectively discover skills from a mixed multi-agent dataset, we utilize an auto-encoder architecture (Kingma & Welling, 2019; Zhang et al., 2023; Liu et al., 2025) to learn skills in an end-to-end manner as shown in Figure 1(b). Specifically, we leverage a global encoder $q_m(z_m^i|s, \boldsymbol{a}, i)$ for each task $m$ to output a skill embedding $z_m^i$ for each agent $i$ based on the global coordination pattern expressed by the state-action pair. Then we decode the skill embeddings $z_m^i$ into actions via an action decoder $\pi_m^i(a^i|\tau^i, z_m^i)$, which is also the skill-conditioned policy. The global encoder $q_m$ and the action decoder $\pi_m^i$ process their inputs via the population invariant network in Equation (4) and are trained via minimizing the policy loss:

$$L_{\pi,q}^{(1)} = -\mathbb{E}_{(\tau^i, a^i) \sim D_m, z_m^i \sim q_m}$$
$$\left[ \exp \left( \frac{w_m^i(\boldsymbol{o})}{\beta_m} A_m^i(o^i, a^i) \right) \cdot \log \pi_m^i(a^i|\tau^i, z_m^i) \right], \quad (5)$$

where $A_m^i(o^i, a^i) = Q_m^i(o^i, a^i) - V_m^i(o^i)$ is the individual advantage function and $\beta_m$ is a hyperparameter.

Moreover, since agents can only access local information during execution, we further introduce a local encoder $p_m(z_m^i|\tau^i)$ that utilizes only the historical trajectory of the individual agent $\tau^i$ to infer the coordination skill $z_m^i$. And we utilize the following KL loss to distill knowledge to help agents to infer $z_m^i$ with local information:

$$L_p = \mathbb{E}_{(s,\tau^i,\boldsymbol{a}) \sim D_m} \left[ D_{KL}(q_m(\cdot|s, \boldsymbol{a}, i) \| p_m(\cdot|\tau^i)) \right]. \quad (6)$$

### 4.2. Skill Partition and Reusability Estimation

As the continual tasks have multi-modal action distribution, designing only one skill encoder would suffer from modal collapse when transferring among tasks. We opt for the multi-head architecture (MH) (Figure 1(b)) due to its simplicity and broad use (Kessler et al., 2022; Wolczyk et al.,

2022; Wołczyk et al., 2021):

$$\pi_m^i(a^i|\tau^i; z_k^i) = G_k^\pi(F(\tau^i, z_k^i)),$$
$$p_m(z_k^i|\tau^i) = G_k^p(F(\tau^i, z_k^i)). \quad (7)$$

The feature extractor $F$ is shared among tasks and is composed of a transformer as in Equation (4), followed by a GRU (Chung et al., 2014) for modeling sequences, or MLP to represent single timesteps. The output heads $G_k^\pi$ and $G_k^p$ are implemented as MLPs, representing action decoders and the prior distribution of skills respectively, for tasks $k = 1, \cdots, m$. Besides, forgetting may occur as the scenario shifts, thus we enforce the following $L_2$ loss on the feature extractor $F$ for task $m \geq 2$:

$$L_F = \lambda_{\text{reg}} \|\theta_{F,m} - \theta_{F,1}\|_2^2, \quad (8)$$

where $\lambda_{\text{reg}}$ is a hyperparameter determining the strength of this loss, $\theta_{F,1}$ is the parameter checkpoint of $F$ saved after training on task 1 and $\theta_{F,m}$ is the parameter of $F$ when training on task $m$.

To reuse the skills acquired previously to guide the discovery of current task, we propose to augment the advantage function with the weighted sum of action logits as follows:

$$\tilde{A}_m^i(s, \boldsymbol{o}, a^i, \{\tilde{z}_k^i\}_{k=1}^m) :=$$
$$\frac{w_m^i(\boldsymbol{o})}{\beta_m} A_m^i(o^i, a^i) + \sum_{k=1}^m \Delta_m^k(s) \log \pi_m^i(a^i|\tau^i, \tilde{z}_k^i), \quad (9)$$

where $\{\tilde{z}_k^i\}_{k=1}^m$ are sampled from the local encoder $p_m(z_k^i|\tau^i)$ for all $k = 1, \cdots, m$, and $\Delta_m^k(s) = \frac{\beta_k d_k(s)}{\sum_{l=1}^m \beta_l d_l(s)}$ is the reusability score calculated via the empirical state distribution $d_k(s)$ estimated from the dataset of task $k$. Intuitively, $\Delta_m^k(s)$ represents the relative confidence of the agent on whether skill $\tilde{z}_k^i$ is reusable and the

logits $\log \pi_m^i(a^i|\tau^i, \tilde{z}_k^i)$ in $\tilde{A}_m^i$ guide the advantage function based on the behavior of the skill $z_k^i$ when observing $\tau^i$. To estimate $d_k(s)$, we consider noise contrastive estimation (NCE) (Gutmann & Hyvärinen, 2012) as an efficient approach. Specifically, let $s^- = s + \mathcal{N}(0, \sigma_s I)$ be the perturbed states and $p(s^-)$ be its distribution, where $\sigma_s > 0$ is a hyperparameter that controls the variance of the noise. We learn a multi-head state density estimator $E_k(s) = G_k^E(F^E(s))$ to approximate the ratio $d_k(s)/p(s^-)$ via minimizing the NCE loss:

$$L_{NCE} = -\,\mathbb{E}_{s \sim D_k}\left[\log \sigma(E_{\eta_k}(s))\right] \tag{10}$$
$$-\,\mathbb{E}_{s^- \sim p}\left[\log \sigma(-E_{\eta_k}(s^-))\right].$$

We also enforce an $L_2$ loss in the same form of Equation (8) on the state feature extractor to prevent forgetting. The skill-augmented policy loss is:

$$L_{\pi,q}^{(2)} = -\mathbb{E}_{(s,\tau^i,a^i) \sim D_m, z_m^i \sim q_m, \{\tilde{z}_k^i\}_{k=1}^m \sim p_m} \tag{11}$$
$$\left[\exp\left(\tilde{A}_m^i(s, \boldsymbol{o}, a^i, \{\tilde{z}_k^i\}_{k=1}^m)\right) \cdot \log \pi_m^i(a^i|\tau^i, z_m^i)\right].$$

Note that a copy of $\pi_m^i$ is created to calculate $\log \pi_m^i$ in $\tilde{A}_m^i$, the gradient is stopped and will not flow back via $\tilde{A}_m^i$. We set $\sigma(x) = \frac{1}{1+e^{-x}}$ in $L_{NCE}$. When facing a new task $m$, we identify if some head $k \in \{1, \cdots, m\}$ can be reused based on the expected state density $\mathbb{E}_{s \sim D_m}[d_k(s)]$. If the expectation is above a prespecified threshold $d_0$, indicating a high probability that the states have been seen before, then head $k$ can be reused for task $m$. Otherwise, the confidence of current heads is low and expansion is needed.

## 4.3. Overall Pipeline

The learning process of COMAD follows the multi-agent task stream as shown in Algorithm 1. For the first task, we allocate output heads $G_1^\pi, G_1^p, G_1^E$ for the action decoder, skill prior and the state density estimator respectively. Then we update the critics $Q, V$ and the mixer $w^i, b$ via the critic losses in Equation (2) and (3), and update the state density estimator $E_1$ via the NCE loss in Equation (10). For the actor, we update the action decoder $\pi_1^i$ and the skill encoder $q_1$ via the vanilla policy loss in Equation (5) and update the local encoder $p_1$ via the KL loss in Equation (6).

Starting from the second task $m \geq 2$, we first evaluate the expected state density $\mathbb{E}_{s \sim D_m}[d_k(s)]$ for all old tasks $k = 1, \cdots, m-1$ and determine to expand new heads $G_m^\pi, G_m^p, G_m^E$ or reuse old heads $G_k^\pi, G_k^p, G_k^E$. The training process of the critic $Q, V$ and mixer $w^i, b$ remains the same, while the objective of state density estimator is added with the $L_2$ loss for the state feature extractor $F^E$. For the actor, the KL loss for the local encoder $p_m$ is the same, while the policy loss proceeds in two stages. In stage 1, the policy is trained via minimizing the vanilla policy loss in Equation

---

**Algorithm 1** COMAD Training

1: **Input:** sequential offline datasets $\mathcal{D} = \{\mathcal{D}_1, \cdots, \mathcal{D}_M\}$
2: **Initialize:** $F, F^E, Q, V, w, b$ and target networks
3: **for** task $m = 1, \cdots, M, \cdots$ **do**
4:     Select head $k^*$ with largest density score that exceeds $d_0$; otherwise allocate a new head.
5:     **for** stage $j = 1, 2$ **do**
6:         **for** batch $B \sim \mathcal{D}_m$ **do**
7:             Update critics and mixer by Eq. (2),(3)
8:             Update local encoder and density estimator by Eq. (6), (10)
9:             **if** $m = 1$ or $j = 1$ **then**
10:                 Set $L_{\pi,q}$ to the vanilla loss in Eq. (5)
11:             **else**
12:                 Set $L_{\pi,q}$ to the augmented loss in Eq. (11)
13:             **end if**
14:             **if** $m \geq 2$ **then**
15:                 Apply $L_2$ regularization
16:             **end if**
17:             Update the actor network and the active heads using $L_{\pi,q}$ and $L_{NCE}$
18:         **end for**
19:     **end for**
20:     **if** $m = 1$ **then**
21:         Save parameters for $L_2$ regularization
22:     **end if**
23: **end for**

---

(5). In stage 2, the policy is trained via minimizing the augmented policy loss in Equation (11) to effectively reuse old skills. Lastly, the $L_2$ loss of the feature extractor $F$ is added to the policy loss in both stages.

During evaluation, we utilize a few samples to approximate $\mathbb{E}_{s \sim D_m}[d_k(s)]$ for all heads and select the most confident one for execution. The full training and evaluation pipeline is presented in Algorithm 2 and 3 in Appendix C.

## 4.4. Theoretical Analysis

Recall that in Section 3 we have defined the skill set $\mathcal{Z}_m$ and we expect the policy $\pi_m^i(a^i|o^i, z_m^i)$ to maximize the performance on the current task and maintain the performance on previous tasks. We formalize this requirement as the following constrained optimization problem:

$$\max_{\pi_m \in \Pi(\mathcal{Z}_m)} \quad J_m(\pi_m) - \beta_m D_{KL}(\pi_m \| \mu_m)$$
$$\text{s.t.} \quad J_k(\pi_m) - \beta_k D_{KL}(\pi_m \| \mu_k) \tag{12}$$
$$\geq J_k(\pi_k) - \beta_k D_{KL}(\pi_k \| \mu_k) - \delta_k,$$
$$\forall k \in \{1, \cdots, m-1\},$$

where $\Pi(\mathcal{Z}_m)$ is the set of all policies corresponding to the skill set $\mathcal{Z}_m$, $J_k(\pi_m)$ is the discounted return of policy $\pi_m$

on task $k = 1, \cdots, m$, $\beta_k$ is a hyperparameter that controls the strength of the KL divergence regularization $D_{KL}(\pi \| \mu)$, $\mu_k$ is the behavior policy corresponding to the dataset of task $k$ and $\delta_k > 0$ is the performance degradation tolerance.

Due to convexity, we can explicitly solve the optimization problem and present the following result:

**Theorem 4.1.** *The optimal solution for problem (12) is*

$$\pi_m^i(a^i | \tau^i, z_m^i) \propto \exp\left( \sum_{k=1}^{m} \Delta_m^{k,*}(s) \log \tilde{\pi}_k^{i,*}(a^i | \tau^i, z_k^i) \right), \tag{13}$$

*where* $\Delta_m^{k,*}(s) = \frac{\lambda_k \beta_k d_k(s)}{\sum_{l=1}^{m} \lambda_l \beta_l d_l(s)}$, $\tilde{\pi}_k^{i,*}(a^i | \tau^i, z_k^i) = \exp\left( \frac{w_k^{i,*}(o)}{\beta_k} A_k^{i,*}(o^i, a^i) + \log \mu_k^i(a^i | \tau^i, z_k^i) \right)$ *is the single task optimal policy,* $\lambda_m = 1$*, and* $\lambda_k, k = 1, \cdots, m-1$ *are Lagrange multipliers satisfying*

$$\begin{aligned} \lambda_k [ & (J_k(\pi_m) - \beta_k D_{KL}(\pi_m \| \mu_k)) \\ & - (J_k(\pi_k) - \beta_k D_{KL}(\pi_k \| \mu_k) - \delta_k)] = 0. \end{aligned} \tag{14}$$

This result indicates that the augmented advantage function in Equation (9) guides the policy to approximate the optimal continual policy in Equation (13), while the reusability score $\Delta_m^k(s)$ approximates $\Delta_m^{k,*}(s)$ that act as a skill gating mechanism. Concretely, it provides us the following insights for the architectural design:

- The Lagrange multipliers $\lambda_1, \cdots, \lambda_m$ act as a gating mechanism for selecting useful skills: they quantify the importance of corresponding task performance constraints and filter out useless skills by recognizing constraint conflicts.

- The reusability scores $\Delta_m^{k,*}(s)$ integrate the multipliers and state densities as fine-grained reusability information: skills should be reused on tasks that are similar (sharing coordination patterns) and on states that are familiar (more confident).

- The optimal solution in Equation (13) presents a unified perspective for comparing and reusing skills across heterogeneous tasks through the geometric mixture of action distributions, which can be generalized to *Bregman Divergences*.

We defer the proof and further discussion to Appendix A.

# 5. Experiments

We evaluate COMAD to answer the following research questions: (1) How will skill partition benefit knowledge reuse? (Section 5.2) (2) What is the continual learning ability of COMAD in different scenarios? (Section 5.3) (3) How are skills transferred across tasks? (Section 5.4) (4) How do different components of COMAD contribute to its performance? (Section 5.5)

## 5.1. Environments, Baselines and Metrics

We evaluate on five diverse environments: (1) Level-based Foraging (LBF) (Papoudakis et al., 2021) is a grid world environment where agents are rewarded to cooperatively collect foods. The positions of foods change among tasks. (2) Cooperative Navigation (CN) (Lowe et al., 2017) is an environment from the multi-agent particle environments (Mordatch & Abbeel, 2018), where agents cooperate to reach landmarks and the number of agents varies among tasks. (3) StarCraft Multiagent Challenge (SMAC) (Samvelyan et al., 2019) focuses on discrete micromanagement with heterogeneous units, based on which we designed two representative scenarios: Marines and Stalker-Zealot. (4) SMACv2 (Ellis et al., 2023) further introduces stochasticity based on SMAC to provide a more challenging benchmark, based on which we design three scenarios: Protoss, Zerg and Terran, based on the races of controllable units. (5) Multiagent MuJoCo (Peng et al., 2021) is a benchmark for continuous multi-agent robotic control. We design two types of scenarios that change in Reward functions or robot Dynamics.

We compare against three categories of methods adapted for offline multi-agent continual cooperation. (1) Offline skill discovery methods: ODIS (Zhang et al., 2023) that extracts skills with a VAE and HiSSD (Liu et al., 2025) that decouples common and task-specific skills via hierarchical VAEs and contrastive learning. Both of them rely on fixed-size skill libraries. (2) Continual learning methods: EWC (Kirkpatrick et al., 2017) prevents forgetting via parameter regularization based on Fisher information, while OWL (Kessler et al., 2022) mitigates interference through parameter isolation using a multi-head architecture. (3) Vanilla methods with single-head architecture: multitask (MT) that trains jointly on all data (as soft upper bound); finetune (FT) that learns each task sequentially (as soft lower bound) and from scratch (FS) that re-initializes for each task (plasticity reference). For all environments and methods, we use a **task-bounded** protocol at both continual training and evaluation, where task identities are only used for output head routing and parameter saving.

Following (Wołczyk et al., 2021), we report three metrics based on the performance $p_k(t) \in [0, 1]$ (win rate in SMAC and SMACv2, or normalized return otherwise) of task $k$ at training step $t$. Each task stream contains $M$ tasks and each task $k$ is trained for $\Delta$ steps during $t \in [(k-1)\Delta, k\Delta]$. The total budget for a task stream is $T = M \cdot \Delta$. Then the three metrics are calculated as follows: (1) Average Performance: $\mathrm{P}(t) := \frac{1}{M} \sum_{k=1}^{M} p_k(t)$, which measures the overall con-

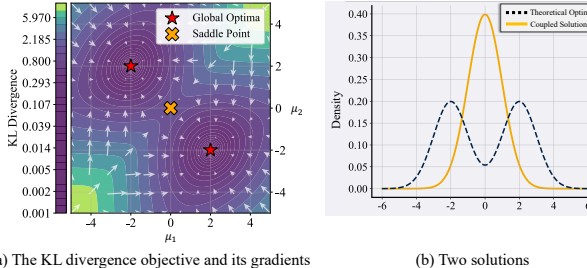

(a) The KL divergence objective and its gradients  (b) Two solutions

*Figure 2.* A toy example of the coupled solution problem. (a) The loss landscape of the objective and its gradients, highlighting two distinct global optima and a saddle point in between. (b) The corresponding density functions of the Coupled Solution that collapses to the saddle point, and the Theoretical Optimal Solution.

tinual learning ability. We report the final performance $P = P(T)$ and performance over time $P(t)$ respectively. (2) Backward Transfer: $BwT_k = p_k(T) - p_k(k\Delta)$, which measures the change of performance between the end of one task and the end of task stream. We report the average backward transfer as $BwT = \frac{1}{M}\sum_{k=1}^{M} BwT_k$. (3) Forward Transfer: $FwT_k = \frac{1}{\Delta}\sum_{t=(k-1)\Delta}^{k\Delta}(p_k(t) - p_k^b(t))$, which measures the fast learning ability of algorithms via the normalized area under their training curves compared to a baseline. FS is chosen as the baseline $p_k^b(t)$ for each task $k$, and we report the average forward transfer as $FwT = \frac{1}{M}\sum_{k=1}^{M} FwT_k$. We defer further details for environments, baselines and metrics to Appendix B.

## 5.2. An Illustrative Example

To show the necessity of the multi-head architecture to handle the multi-modal action data, we conduct a simple numerical experiment based on the following setting:

$$p_{\text{data}} \propto \mathcal{N}(-2, 1) + \mathcal{N}(2, 1)$$
$$p_{\text{model}} \propto \mathcal{N}(\mu_1, 1) + \mathcal{N}(\mu_2, 1) \quad (15)$$
$$d(\mu_1, \mu_2) = D_{KL}(p_{\text{data}} \| p_{\text{model}}),$$

where $p_{\text{data}}$ is the data distribution with two Gaussian modalities, $p_{\text{model}}$ is a model with parameters $\mu_1, \mu_2$, and $d(\mu_1, \mu_2)$ is the optimization objective and is visualized in Figure 2(a) together with its gradients. We observe that: 1) When setting only one degree of freedom, i.e. setting $\mu_1 = \mu_2$, the model converges to the saddle point on the diagonal that corresponds to the coupled solution in Figure 2(b), presenting a geometric mixture of the two modalities in $p_{\text{data}}$; 2) When $\mu_1, \mu_2$ are free, the model can easily converge to the optimal solutions at $(-2, 2)$ or $(2, -2)$ due to convexity, corresponding to the optimal solution shown in Figure 2(b).

The observations suggest that a skill-conditioned policy with sufficient representation ability is the key to partitioning different skills from sequential multi-agent datasets. Without such explicit partitioning, the policy is forced to compress the disparate behavioral modes into a single unimodal rep-

resentation, leading to a failure in distinguishing between heterogeneous coordination patterns. This outcome reflects an interfered policy that conflates conflicting gradients, exactly corresponding to the case where skills are not gated by the coefficient $\Delta_m^{k,*}(s)$ in Theorem 4.1.

## 5.3. Main Results

To evaluate the continual learning capabilities of COMAD, we report the average performance ($P \times 100$) across 9 task streams in Table 1, with representative learning curves shown in Figure 3. COMAD achieves superior overall performance (64.34), significantly outperforming all baselines across diverse scenarios.

The LBF environment, which requires agents to **memorize** distinct goals (food locations), highlights the stability-plasticity dilemma. Firstly, skill discovery methods (ODIS, HiSSD) fail to learn feasible policies due to a lack of plasticity resulting from their fixed-size skill libraries. Meanwhile, monolithic approaches (MT, FS, FT, EWC) suffer from severe catastrophic forgetting, which suggests possible task conflict. In contrast, methods with explicit task separation (OWL) or replay (Rehearsal) succeed by mitigating task conflict. Importantly, COMAD effectively expands its skill library to memorize diverse goals, achieving the best performance on the expert datasets and remains competitive with the best baseline on medium datasets. In addition, the high similarity among CN tasks emphasizes the potential for positive transfer. Consequently, simple baselines like FS, FT and EWC perform well, particularly on medium datasets where fine-tuning is sufficient. However, skill discovery baselines overfit to early, simple tasks and struggle to scale, while MT and Rehearsal are relatively limited by potential task conflicts and training instability. Instead, COMAD leverages this similarity through its reuse mechanism, significantly outperforming all methods on expert datasets by effectively transferring coordination patterns across varying team sizes.

Furthermore, SMAC tasks present a mixed challenge requiring both cross-task transfer and separation, as the task streams are divided into two parts: symmetric tasks (such as 4m and 2s3z) and asymmetric tasks (such as 5m_vs_6m and 2s2z_vs_4z). Consequently, skill discovery baselines (ODIS, HiSSD) suffer from plasticity loss and fail to learn the asymmetric tasks as shown in Figure 3(a). Conventional baselines (FT, FS, EWC) also struggle with this task heterogeneity, often failing to yield feasible policies. Although OWL mitigates task conflicts through isolation, it lacks the transfer capability to exploit shared structures. COMAD instead excels by partitioning heterogeneous behaviors while actively reusing skills, achieving near-optimal performance. This advantage is amplified in SMACv2, where stochasticity and large team scales (up to 20 agents) exacerbate

*Table 1.* Average performance $\pm$ std of different algorithms on task streams from LBF, CN, SMAC, SMACv2 and MAMuJoCo environments. The **best** and second best algorithms are marked and algorithms whose average performances lie within 1 std of the best algorithms are marked with asterisks (*). All results are based on 5 distinct seeds and 32 episodes per seed on each evaluation step. "Overall" reports the average performance of each algorithm over all task streams.

| Task Stream | Dataset | COMAD(ours) | MT | FS | FT | EWC | OWL | Rehearsal | ODIS-FT | HiSSD-FT |
|---|---|---|---|---|---|---|---|---|---|---|
| LBF | Expert | **99.26 ± 0.08** | 79.26 ± 0.14 | 20.00 ± 0.00 | 7.83 ± 6.72 | 13.30 ± 0.53 | 98.65 ± 0.35 | 97.07 ± 0.66 | 16.48 ± 2.86 | 19.17 ± 1.70 |
| | Medium | 65.08 ± 1.34 | 48.49 ± 1.72 | 16.14 ± 0.71 | 7.86 ± 4.00 | 7.45 ± 3.26 | 37.83 ± 13.78 | **66.79 ± 1.15** | 7.06 ± 0.71 | 7.37 ± 1.26 |
| CN | Expert | **78.49 ± 0.41** | 69.60 ± 2.06 | 69.35 ± 0.93 | 69.10 ± 1.84 | 67.18 ± 3.71 | 64.59 ± 1.74 | 72.60 ± 1.01 | 67.26 ± 0.75 | 63.23 ± 2.16 |
| | Medium | 47.05 ± 2.03 | 32.33 ± 3.38 | 59.51 ± 0.60 | **60.67 ± 1.04** | 55.18 ± 1.43 | 33.33 ± 2.21 | | 10.48 ± 1.25 | 12.06 ± 2.01 |
| Marines | Expert | **94.05 ± 0.83** | 48.63 ± 0.49 | 53.84 ± 4.81 | 53.66 ± 4.91 | 58.84 ± 5.08 | 92.52 ± 0.97 | 42.45 ± 28.24 | 17.83 ± 0.70 | 14.51 ± 0.20 |
| | Medium | **55.54 ± 0.66** | 34.63 ± 1.46 | 51.50 ± 3.37 | 18.15 ± 17.84 | 36.91 ± 3.48 | 48.19 ± 2.41 | 1.10 ± 0.69 | 31.79 ± 4.23 | 35.06 ± 1.61 |
| Stalker-Zealot | Expert | 72.20 ± 0.77 | 42.66 ± 0.99 | 9.90 ± 0.46 | 10.38 ± 3.72 | 14.84 ± 1.42 | **77.49 ± 0.29** | 71.40 ± 0.99 | 17.83 ± 1.07 | 27.61 ± 0.82 |
| | Medium | 45.20 ± 1.89 | **52.96 ± 1.08** | 10.74 ± 1.37 | 9.75 ± 2.00 | 16.65 ± 7.67 | 52.54 ± 3.25* | 23.45 ± 22.37 | 35.13 ± 0.97 | 31.07 ± 3.45 |
| Protoss | Medium | **59.77 ± 1.76** | 47.40 ± 0.73 | 24.25 ± 0.82 | 21.00 ± 1.14 | 23.56 ± 1.17 | 57.05 ± 0.28 | 44.90 ± 1.27 | 8.39 ± 0.99 | 18.04 ± 2.71 |
| Zerg | Medium | **59.98 ± 2.31** | 11.97 ± 2.20 | 34.20 ± 1.49 | 35.92 ± 0.22 | 35.70 ± 1.32 | 56.89 ± 1.45 | 22.10 ± 21.18 | 4.22 ± 2.99 | 9.77 ± 9.10 |
| Terran | Medium | **57.35 ± 1.60** | 44.74 ± 1.03 | 37.54 ± 1.20 | 35.37 ± 2.13 | 38.82 ± 1.24 | 55.09 ± 1.34 | 24.21 ± 7.78 | 10.37 ± 8.11 | 4.66 ± 1.74 |
| Reward | Expert | **77.75 ± 4.64** | 63.73 ± 5.93 | 10.18 ± 1.30 | 5.09 ± 1.06 | 15.45 ± 1.56 | 75.46 ± 5.08* | 5.33 ± 3.73 | 8.20 ± 1.76 | 2.55 ± 0.47 |
| | Medium | **62.16 ± 3.22** | 30.24 ± 5.35 | 11.55 ± 2.06 | 9.01 ± 0.22 | 10.58 ± 1.00 | 58.15 ± 3.10 | 11.53 ± 2.13 | 11.23 ± 0.64 | 11.95 ± 1.17 |
| Dynamics | Expert | 46.66 ± 0.96 | **55.28 ± 4.27** | 23.24 ± 1.53 | 19.48 ± 0.53 | 15.65 ± 1.23 | 46.94 ± 4.31 | 14.77 ± 1.37 | 14.56 ± 1.52 | 16.86 ± 1.30 |
| | Medium | 44.54 ± 3.73 | **53.14 ± 5.53** | 19.52 ± 1.05 | 15.72 ± 0.23 | 16.54 ± 0.63 | 42.87 ± 2.45 | 16.39 ± 3.78 | 16.03 ± 1.81 | 16.70 ± 1.27 |
| Overall | | **64.34** | 47.67 | 30.10 | 25.26 | 28.44 | 59.84 | 36.93 | 18.46 | 19.37 |

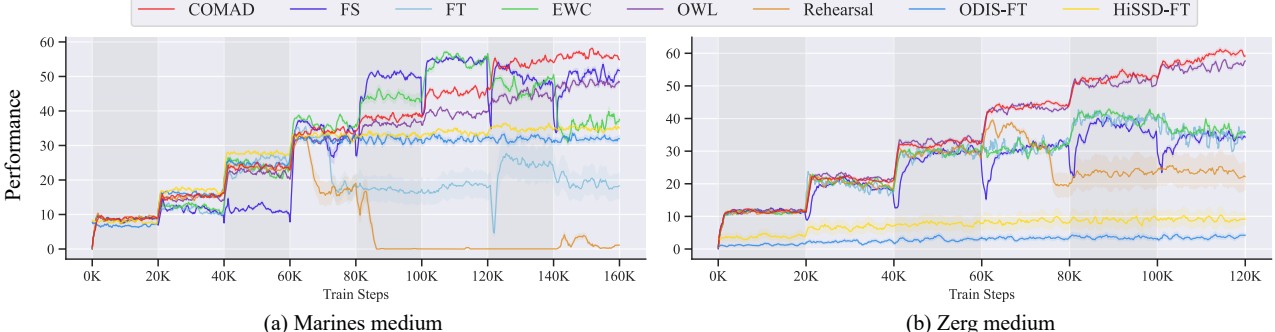

(a) Marines medium      (b) Zerg medium

*Figure 3.* Representative learning curves of all method except MT, where the shaded areas correspond to $0.2 \times$ std for visual clarity. The training timeline is segmented into 8 (left) or 6 (right) 20K-step intervals, which represent the training phases of tasks in corresponding task streams. The value at each point represents the average performance evaluated across the current task and all encountered tasks.

the problem of the evolving skill space. As shown in Figure 3(b), COMAD significantly outperforms baselines that suffer from plasticity loss (ODIS, HiSSD) or potential interference (Rehearsal).

Lastly, Reward tasks in MAMuJoCo further reinforce the importance of skill memorization presented in LBF. Results show that COMAD and OWL dominate while Rehearsal and other baselines struggle due to the complexity of the continuous space. Similarly, in Dynamics tasks, extreme physical changes render most continual methods sub-optimal compared to MT. Nevertheless, COMAD remains the most competitive continual approach, demonstrating its robustness even under severe dynamic shifts. We defer complete results and further discussion to Appendix D.

### 5.4. Skill Transfer Analysis

To understand the skill transfer process in COMAD, we analyze a representative slice of the learning curve on the Marines-expert task, as shown in Figure 4. Concretely, we

zoom in on the $80K \sim 120K$ training steps, which covers the learning of 5m_vs_6m and 7m_vs_8m.

During stage 1 of 5m_vs_6m, the agent acquires the Focus Fire skill to coordinate units to target specific enemies, achieving a win rate about 0.6. Meanwhile, the output head of 5m_vs_6m is reused with a high confidence score for the evaluation of 7m_vs_8m, which improves the win rate to near 0.3 on task 7m_vs_8m, showcasing zero-shot generalization. Furthermore, during stage 2, the skill-augmented objective effectively integrates skills acquired from earlier tasks and improves the win rate to exceed 0.4, providing a higher starting point and a base for faster learning on task 7m_vs_8m, evidencing the existence of forward transfer.

Additionally, when learning 7m_vs_8m, the win rate of 5m_vs_6m improves from 0.6 to roughly 0.8. Skill visualization reveals the mechanism behind such positive backward transfer: while learning the novel Positioning skill for 7m_vs_8m, the agent also finetunes its Disperse skill for 5m_vs_6m through the shared feature extractor. The

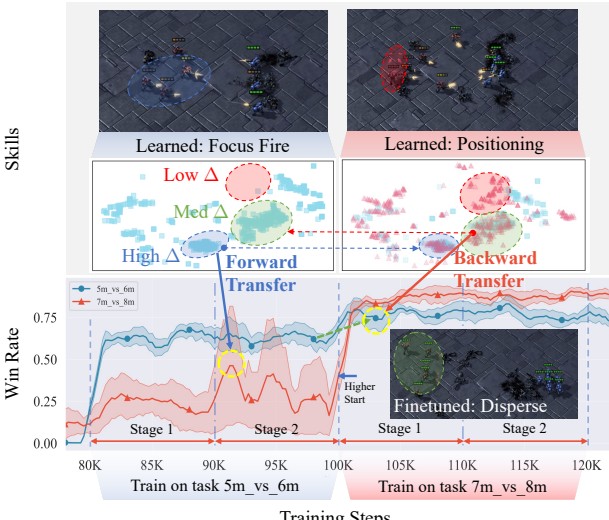

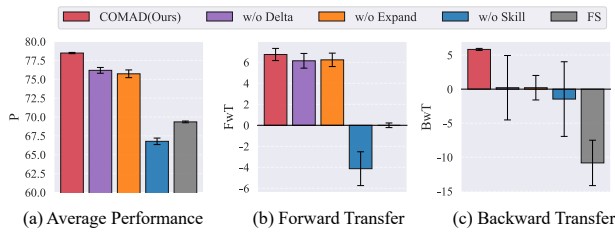

*Figure 4.* A slice of the learning curve and skill visualization on 5m_vs_6m and 7m_vs_8m from the Marines-expert task stream. The lower row presents the win rates across different training stages for each task, the middle row visualizes the evolution of skill embeddings and their relative reusability, and the upper row shows the outcomes of the learned skills.

*Figure 5.* Metrics of ablation studies on CN-expert task.

remarkable phenomenon showcases the backward transfer ability of COMAD through skill partitioning.

### 5.5. Ablation Study

To dissect the contributions of each component within CO-MAD, we conduct ablation studies on the CN-expert task, as depicted in Figure 5. Firstly, removing the skill encoders (w/o Skill) affects both learning and bidirectional transfer the most, emphasizing the necessity of partitioning skills from data. Besides, disabling skill library expansion (w/o Expand) and uniform reusability estimation (w/o Delta) also downgrade the performance and transfer, especially the backward transfer. Furthermore, ablation studies on Marines-expert task show that partitioning skills with MH is essential for learning and transfer, while reusability estimation has major influence on forward transfer (in Appendix D.1). We also analyze the sensitivity of the noise scale $\sigma_s$ and the confidence threshold $d_0$, showcasing that moderate choices of $\sigma_s$ and $d_0$ maximize performance and transfer. Additional results are detailed in Appendix D.2.

### 6. Final Remarks

In this work, we propose a principled framework for **C**ontinual **O**ffline **M**ulti-**a**gent Skill **D**iscovery via Skill Partition and Reuse (COMAD), to handle the exponential growth of coordination skill space in offline multi-agent task streams. COMAD discovers skills with a VAE-based skill encoder, actively expands its skill library and reuses them based on reusability estimation and a skill-augmented objective, thereby effectively promoting skill learning and transfer. Theoretical results indicate that COMAD approximates the optimal continual policy, and empirical results demonstrate its effectiveness across diverse environments. Overall, COMAD contributes a new multi-agent continual learning formulation, a reuse mechanism via skill guidance and corresponding theoretical analysis compared to previous works. However, relying on state density to approximate the skill gating mechanism may be insufficient for complex real-world scenarios, where skills should be reused based on fine-grained semantic reusability. Future work could explore improving this gating mechanism by integrating Large Language Models (LLMs) or human feedback.

### Acknowledgements

This work was supported by the National Natural Science Foundation of China under Grants 62506159, U24A20324, the Natural Science Foundation of Jiangsu under Grants BK20241199, BK20243039, the "111 Center" (No. B26023), and Fundamental and Interdisciplinary Disciplines Breakthrough Plan of the Ministry of Education of China (No. JYB2025XDXM118). We thank the anonymous reviewers for their support and helpful discussions on improving the paper.

### Impact Statement

The work presented in this paper aims to enhance the adaptability and sample efficiency of offline multi-agent reinforcement learning methods in continual task stream settings. The proposed framework provides an efficient approach for future research on more sustainable and versatile autonomous multi-agent systems. We declare that this work does not involve any ethical issues, and thus no special discussion on ethical issues is required.

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

# A. Theoretical Results

## A.1. Proof for Theorem 4.1

We first introduce the necessary assumption and lemmas regarding the multi-agent policy factorization, and then derive the optimal continual learning policy. To bridge the gap between the joint policy optimization and independent agent execution, we adopt the skill-based factorization assumption:

**Assumption A.1** (Behavior Factorization). The joint behavior policy $\mu_m^{tot}(\boldsymbol{a}|\boldsymbol{o})$ of task $m$ is skill-conditioned, i.e. in the form of $\mu_m^{tot}(\boldsymbol{a}|\boldsymbol{o}, \boldsymbol{z}_m)$, and it can be factorized into the product of individual policies of $N$ agents:

$$\mu_m^{tot}(\boldsymbol{a}|\boldsymbol{o}) = \mu_m^{tot}(\boldsymbol{a}|\boldsymbol{o}, \boldsymbol{z}_m) = \prod_{i=1}^{N} \mu_m^i(a^i|o^i, z_m^i), \tag{16}$$

where $\boldsymbol{z}_m = \{z_m^1, \ldots, z_m^N\}$ corresponds to the set of skills for all agents.

To decompose the global optimal policy, we restate the following results from (Wang et al., 2023):

**Lemma A.2** (Restated Proposition 4.2 of (Wang et al., 2023)). *Assume that the skill-conditioned policy set $\Pi(\mathcal{Z}_m)$ is expressive enough such that $\Pi(\mathcal{Z}_m) = \Pi$, i.e., $\Pi(\mathcal{Z}_m)$ can represent all possible policies. Then for a behavior-regularized Dec-POMDP with reverse KL regularization, the **single task** optimal global policy $\tilde{\pi}_m^{tot,*}$ for task $m$ satisfies:*

$$\tilde{\pi}_m^{tot,*}(\boldsymbol{a}|\boldsymbol{o}, \boldsymbol{z}_m) = \mu_m^{tot}(\boldsymbol{a}|\boldsymbol{o}, \boldsymbol{z}_m) \cdot \exp\left(\frac{Q_m^{tot,*}(\boldsymbol{o}, \boldsymbol{a}) - V_m^{tot,*}(\boldsymbol{o})}{\beta_m}\right), \tag{17}$$

*where $Q_m^{tot,*}$ and $V_m^{tot,*}$ are optimal value functions.*

Following OMIGA (Wang et al., 2023), incorporating the value decomposition scheme in Equation (1) and behavior decomposition in Equation (16) into the optimal global policy $\tilde{\pi}^{tot,*}$ in Equation (17) naturally leads to the decomposition of the optimal global policy:

**Lemma A.3** (Equivalence of Decompositions). *Under the value decomposition scheme in Equation (1) and behavior policy decomposition in (16), we have $\tilde{\pi}_m^{tot,*}(\boldsymbol{a}|\boldsymbol{o}, \boldsymbol{z}_m) = \prod_{i=1}^{n} \tilde{\pi}_m^{i,*}(a^i|o^i, z_m^i)$, where $\tilde{\pi}_m^{i,*}$ is defined as follows:*

$$\tilde{\pi}_m^{i,*}(a^i|o^i, z_m^i) = \mu_m^i(a^i|o^i, z_m^i) \cdot \exp\left(\frac{w_m^{i,*}(\boldsymbol{o})}{\beta_m} A_m^{i,*}(o^i, a^i)\right), \tag{18}$$

*where $A_m^{i,*}(o^i, a^i) = Q_m^{i,*}(o^i, a^i) - V_m^{i,*}(o^i)$ is the individual advantage function. Conversely, the decomposition of $\tilde{\pi}_m^{tot,*}$ also implies the decomposition of $\mu_m^{tot}$ under the same condition.*

*Proof.* Substituting the factorized behavior policy into the optimal policy form:

$$\begin{aligned}
\tilde{\pi}_m^{tot,*}(\boldsymbol{a}|\boldsymbol{o}, \boldsymbol{z}_m) &= \mu_m^{tot}(\boldsymbol{a}|\boldsymbol{o}, \boldsymbol{z}_m) \cdot \exp\left(\frac{Q_m^{tot,*}(\boldsymbol{o}, \boldsymbol{a}) - V_m^{tot,*}(\boldsymbol{o})}{\beta_m}\right) \\
&= \left(\prod_{i=1}^{N} \mu_m^i(a^i|o^i, z_m^i)\right) \exp\left(\sum_{i=1}^{N} \frac{w_m^{i,*}(\boldsymbol{o})}{\beta_m} A_m^{i,*}(o^i, a^i)\right) \\
&= \prod_{i=1}^{N} \left[\mu_m^i(a^i|o^i, z_m^i) \exp\left(\frac{w_m^{i,*}(\boldsymbol{o})}{\beta_m} A_m^{i,*}(o^i, a^i)\right)\right] \\
&= \prod_{i=1}^{N} \tilde{\pi}_m^{i,*}(a^i|o^i, z_m^i).
\end{aligned} \tag{19}$$

On the other hand, rewriting the deduction reversely shows that the decomposition of $\tilde{\pi}_m^{tot,*}$ also implies the decomposition of $\mu_m^{tot}$. To sum up, the decompositions of $\mu_m^{tot}$ and $\tilde{\pi}_m^{tot,*}$ are equivalent. $\square$

Based on the above results, we show that the optimal solution of the continual learning problem (12) has the following form:

$$\pi_m^i(a^i|o^i, z_m^i) \propto \exp\left(\sum_{k=1}^{m} \Delta_m^{k,*}(s) \log \tilde{\pi}_k^{i,*}(a^i|o^i, z_k^i)\right), \tag{20}$$

where $\Delta_m^{k,*}(s) = \frac{\lambda_k \beta_k d_k(s)}{\sum_{l=1}^{m} \lambda_l \beta_l d_l(s)}$, $\tilde{\pi}_k^{i,*}(a^i|o^i, z_k^i) = \exp\left(\frac{w_k^{i,*}(o)}{\beta_k} A_k^{i,*}(o^i, a^i) + \log \mu_k^i(a^i|o^i, z_k^i)\right)$ is the single task optimal policy, $\lambda_m = 1$, and $\lambda_k, k = 1, \cdots, m-1$ are Lagrange multipliers satisfying

$$\lambda_k[(J_k(\pi_m) - \beta_k D_{KL}(\pi_m \| \mu_k)) \\ - (J_k(\pi_k) - \beta_k D_{KL}(\pi_k \| \mu_k) - \delta_k)] = 0. \tag{21}$$

*Proof.* We restate the continual learning problem in Equation (12) formally as follows:

$$\begin{aligned} \max_{\pi_m^{tot}} \quad & \mathbb{E}_{P_m, \pi_m^{tot}}[Q_m^{tot}(o, a)] - \beta_m D_{KL}(\pi_m^{tot} \| \mu_m^{tot}) \\ \text{s.t.} \quad & \mathbb{E}_{P_k, \pi_m^{tot}}[Q_k^{tot}(o, a)] - \beta_k D_{KL}(\pi_m^{tot} \| \mu_k^{tot}) \geq L_k, \quad \forall k \in \{1, \ldots, m-1\}, \end{aligned} \tag{22}$$

where $L_k := J_k(\pi_k) - \beta_k D_{KL}(\pi_k \| \mu_k) - \delta_k$ represents the performance lower bound for task $k$.

Let $\lambda_k \geq 0$ be the Lagrange multipliers for the performance constraints. For notational uniformity, we set $\lambda_m = 1$ and $L_m = 0$ for the objective of current task. Then the Lagrangian function $\mathcal{L}$ is:

$$\begin{aligned} \mathcal{L}(\pi_m^{tot}, \boldsymbol{\lambda}, u_m, \eta_m) = \sum_{k=1}^{m} \lambda_k &\left[ \sum_{o} d_k^{\pi_m^{tot}}(o) \sum_{a} \pi_m^{tot}(a|o, z_m) \left( Q_k^{tot}(o, a) - \beta_k \log \frac{\pi_m^{tot}(a|o, z_m)}{\mu_k^{tot}(a|o, z_k)} \right) - L_k \right] \\ &- \sum_{o} d_m^{\pi_m^{tot}}(o) \left[ u_m(o) \left( \sum_{a} \pi_m^{tot}(a|o, z_m) - 1 \right) - \sum_{a} \eta_m(a|o, z_m) \pi_m^{tot}(a|o, z_m)) \right], \end{aligned} \tag{23}$$

where $u_m(o)$ is the multiplier ensuring the normalization constraint $\sum_{a} \pi_m^{tot}(a|o, z_m) = 1$, and $\eta_m(a|o, z_m)$ is the multiplier ensuring the non-negativity constraint $\pi_m^{tot}(a|o, z_m) \geq 0$. Notably, $d_k^{\pi_m^{tot}}(o)$ is the stationary joint observation distribution of the current global policy $\pi_m^{tot}$ on the $k$-th Dec-POMDP. However, it can be difficult to evaluate under the offline continual learning setting. Inspired by Stationary Distribution Correction Estimation (DICE) literature (Lee et al., 2021), we replace this marginal joint observation distribution $d_k^{\pi_m^{tot}}(o)$ with the empirical distribution $d_k(o) := d^{D_k}(o)$, i.e., the joint observation induced by dataset $D_k$.

According to the Karush-Kuhn-Tucker (KKT) conditions, taking the functional derivative of $\mathcal{L}$ with respect to $\pi_m^{tot}(a|o, z_m)$ and setting it to zero, we have:

$$\begin{aligned} &\sum_{k=1}^{m} \lambda_k d_k(o) \left( Q_k^{tot}(o, a) - \beta_k \log \pi_m^{tot}(a|o, z_m) - \beta_k + \beta_k \log \mu_k^{tot}(a|o, z_k) \right) \\ &\quad - d_m(o) u_m(o) + d_m(o) \eta_m(a|o, z_m) = 0, \end{aligned} \tag{24}$$

$$\sum_{a} \pi_m^{tot}(a|o, z_m) = 1,$$

$$\eta_m(a|o, z_m) \pi_m^{tot}(a|o, z_m) = 0,$$

$$0 \leq \pi_m^{tot}(a|o, z_m) \leq 1 \text{ and } 0 \leq \eta_m(a|o, z_m),$$

$$\lambda_k \left[ \mathbb{E}_{\pi_m^{tot}}[Q_k^{tot}(o, a)] - \beta_k D_{KL}(\pi_m^{tot} \| \mu_k^{tot}) - L_k \right] = 0, \forall k \in \{1, \cdots, m-1\}. \tag{25}$$

Rearranging terms in Equation (24) to solve for $\pi_m^{tot}(a|o, z_m)$:

$$\begin{aligned} &\left( \sum_{k=1}^{m} \lambda_k \beta_k d_k(o) \right) \log \pi_m^{tot}(a|o, z_m) \\ &= \sum_{k=1}^{m} \lambda_k \beta_k d_k(o) \left( Q_k^{tot}(o, a)/\beta_k + \log \mu_k^{tot}(a|o, z_m) \right) + C(o) + d_m(o) \eta_m(a|o, z_m), \end{aligned} \tag{26}$$

where $C(\boldsymbol{o})$ absorbs terms independent of global action $\boldsymbol{a}$. Next, we define the Reusability Score $\Delta_m^{k,*}(\boldsymbol{o})$ as:

$$\Delta_m^{k,*}(\boldsymbol{o}) = \frac{\lambda_k \beta_k d_k(\boldsymbol{o})}{\sum_{j=1}^m \lambda_j \beta_j d_j(\boldsymbol{o})}. \tag{27}$$

Note that $\sum_{k=1}^m \Delta_m^{k,*}(\boldsymbol{o}) = 1$. Further rearranging terms in Equation (26) and omitting $C(\boldsymbol{o})$ terms, the optimal joint policy becomes:

$$\begin{aligned}
\pi_m^{tot,*}(\boldsymbol{a}|\boldsymbol{o}, \boldsymbol{z}_m) &\propto \exp\left(\sum_{k=1}^m \Delta_m^{k,*}(\boldsymbol{o})\left(\frac{A_k^{tot}(\boldsymbol{o}, \boldsymbol{a})}{\beta_k} + \log \mu_k^{tot}(\boldsymbol{a}|\boldsymbol{o}, \boldsymbol{z}_k)\right) + \frac{d_m}{\sum_{j=1}^m \lambda_j \beta_j d_j(\boldsymbol{o})}\eta_m(\boldsymbol{a}|\boldsymbol{o}, \boldsymbol{z}_m)\right) \\
&= \exp\left(\sum_{k=1}^m \Delta_m^{k,*}(\boldsymbol{o})\log \tilde{\pi}_k^{tot,*}(\boldsymbol{a}|\boldsymbol{o}, \boldsymbol{z}_k) + \frac{d_m}{\sum_{j=1}^m \lambda_j \beta_j d_j(\boldsymbol{o})}\eta_m(\boldsymbol{a}|\boldsymbol{o}, \boldsymbol{z}_m)\right).
\end{aligned} \tag{28}$$

Since $\pi_m^{tot,*} = \exp(\cdots) > 0$, the complementary slackness condition $\eta_m(\boldsymbol{a}|\boldsymbol{o}, \boldsymbol{z}_m)\pi_m^{tot}(\boldsymbol{a}|\boldsymbol{o}, \boldsymbol{z}_m) = 0$ enforces $\eta_m(\boldsymbol{a}|\boldsymbol{o}, \boldsymbol{z}_m) = 0$ for all $\boldsymbol{o}, \boldsymbol{a}, \boldsymbol{z}_m$, thus we have:

$$\pi_m^{tot,*}(\boldsymbol{a}|\boldsymbol{o}, \boldsymbol{z}_m) \propto \exp\left(\sum_{k=1}^m \Delta_m^{k,*}(\boldsymbol{o})\log \tilde{\pi}_k^{tot,*}(\boldsymbol{a}|\boldsymbol{o}, \boldsymbol{z}_k)\right). \tag{29}$$

To derive the optimal solution for individual policies, we apply Lemma A.3 to further decompose $\pi_m^{tot,*}$:

$$\begin{aligned}
\log \pi_m^{tot,*}(\boldsymbol{a}|\boldsymbol{o}, \boldsymbol{z}_m) &= \sum_{k=1}^m \Delta_m^{k,*}(\boldsymbol{o})\log \tilde{\pi}_k^{tot,*}(\boldsymbol{a}|\boldsymbol{o}, \boldsymbol{z}_k) + \log C(\boldsymbol{o}) \\
&= \sum_{k=1}^m \Delta_m^{k,*}(\boldsymbol{o})\sum_{i=1}^N \log \tilde{\pi}_k^{i,*}(a^i|o^i, z_k^i) + \log C(\boldsymbol{o}) \\
&= \sum_{i=1}^N \sum_{k=1}^m \Delta_m^{k,*}(\boldsymbol{o})\log \tilde{\pi}_k^{i,*}(a^i|o^i, z_k^i) + \log C(\boldsymbol{o})
\end{aligned} \tag{30}$$

Based on the equivalence of the decompositions of $\mu_m^{tot}$ and $\tilde{\pi}_m^{tot,*}$, matching terms for each agent $i$ yields the result:

$$\log \pi_m^{i,*}(a^i|o^i, z_k^i) = \sum_{k=1}^m \Delta_m^{k,*}(\boldsymbol{o})\log \tilde{\pi}_k^{i,*}(a^i|o^i, z_k^i) + \text{const.} \tag{31}$$

Gathering Equation (25) and Equation (31), we conclude the proof. $\qquad \square$

## A.2. Derivation of Policy Learning Objectives

In this section, we derive the tractable learning objectives used to train the skill-conditioned policy $\pi_\theta$ by explicitly connecting the theoretical optimal policies derived in Theorem 4.1 to the practical loss functions. We adopt the following settings and approximations for practical instantiation:

1. **Parameters:** We set the parameters $\beta_k, k = 1, \cdots, m$ to a fixed constant $\beta_1 = \cdots = \beta_m = \alpha = 10$.

2. **Trajectory-based Policy:** To handle partial observability, we replace all the observation-conditioned policies with trajectory-conditioned policies, e.g., $\tilde{\pi}_k^{i,*}(a^i|\tau^i, z_k^i)$, which is approximated by a parameterized multi-head model $\pi_\theta(a^i|\tau^i, z_k^i)$ of the same form in Equation (7).

3. **Centralized Reusability:** We approximate the reusability scores using global state information instead of joint observations during centralized training: $\Delta_m^{k,*}(\boldsymbol{o}) \approx \Delta_m^{k,*}(s)$.

**Derivation of $L_{\pi,q}^{(1)}$ in Equation** (5)    First, we consider the basic skill discovery objective within single task. The optimal policy form is given by the advantage-weighted behavior policy:

$$\tilde{\pi}_m^i(a^i|\tau^i, z_m^i) = \mu_m^i(a^i|\tau^i, z_m^i)\exp\left(\frac{w_m^i(\boldsymbol{o})}{\beta_m}A_m^i(o^i, a^i)\right). \tag{32}$$

We aim to minimize the KL divergence between this optimal target $\tilde{\pi}_m^i$ and our parameterized policy $\pi_\theta$:

$$\min_\theta \mathbb{E}_{(\tau^i, a^i)\sim\mathcal{D}_m, z_m^i\sim q_m}\left[D_{KL}(\tilde{\pi}_m^i(\cdot|\tau^i, z_m^i)\|\pi_\theta(\cdot|\tau^i, z_m^i))\right]. \tag{33}$$

Expanding the KL divergence:

$$\begin{aligned}
D_{KL}(\tilde{\pi}_m^i(\cdot|\tau^i, z_m^i)\|\pi_\theta(\cdot|\tau^i, z_m^i)) &= \sum_{a^i}\tilde{\pi}_m^i(a^i|\tau^i, z_m^i)\log\frac{\tilde{\pi}_m^i(a^i|\tau^i, z_m^i)}{\pi_\theta(a^i|\tau^i, z_m^i)} \\
&= -\sum_{a^i}\tilde{\pi}_m^i(a^i|\tau^i, z_m^i)\log\pi_\theta(a^i|\tau^i, z_m^i) + \text{const.}
\end{aligned} \tag{34}$$

Substituting $\tilde{\pi}_m^i$, we can rewrite the expectation over actions using importance sampling relative to the behavior policy $\mu_m^i$. Since $\tilde{\pi}_m^i = \mu_m^i\exp(w_m^i A_m^i/\beta_m)$, the importance weight becomes the exponential advantage:

$$\begin{aligned}
&\mathbb{E}_{(\tau^i, a^i)\sim\mathcal{D}_m, z_m^i\sim q_m}\left[D_{KL}(\tilde{\pi}_m^i(\cdot|\tau^i, z_m^i)\|\pi_\theta(\cdot|\tau^i, z_m^i))\right] \\
&= -\mathbb{E}_{(\tau^i, a^i)\sim\mathcal{D}_m, z_m^i\sim q_m}\left[\frac{\tilde{\pi}_m^i(a^i|\tau^i, z_m^i)}{\mu_m^i(a^i|\tau^i, z_m^i)}\cdot\log\pi_\theta(a^i|\tau^i, z_m^i)\right] + \text{const} \\
&= -\mathbb{E}_{(\tau^i, a^i)\sim\mathcal{D}_m, z_m^i\sim q_m}\left[\exp\left(\frac{w_m^i(\boldsymbol{o})}{\beta_m}A_m^i(o^i, a^i)\right)\cdot\log\pi_\theta(a^i|\tau^i, z_m^i)\right] + \text{const.} \\
&= L_{\pi,q}^{(1)}(\theta) + \text{const.}
\end{aligned} \tag{35}$$

**Derivation of $L_{\pi,q}^{(2)}$ in Equation** (11)    For the skill reuse phase, the target policy $\pi_m^{i,*}$ is the geometric weighted average of single task policies (Theorem 4.1). To facilitate in-sample learning, we approximate the ratio between the optimal policy and the behavior policy as follows:

$$\begin{aligned}
\frac{\pi_m^{i,*}(a^i|\tau^i, z_m^i)}{\mu_m^i(a^i|\tau^i, z_m^i)} &\propto \frac{\exp\left(\sum_{k=1}^m\Delta_m^{k,*}(s)\log\tilde{\pi}_k^{i,*}(a^i|\tau^i, z_k^i)\right)}{\mu_m^i(a^i|\tau^i, z_m^i)} \\
&\approx \exp\left(\frac{w_m^i(o)}{\beta_m}A_m^i(o^i, a^i) + \sum_{k=1}^m\Delta_m^{k,*}(s)\log\tilde{\pi}_k^{i,*}(a^i|\tau^i, z_k^i)\right).
\end{aligned} \tag{36}$$

We define the Augmented Advantage $\tilde{A}_m^i$ to capture both the new task advantage and the knowledge of skills:

$$\tilde{A}_m^i(s, \boldsymbol{o}, a^i, \{z_k^i\}_{k=1}^m) = \frac{w_m^i(\boldsymbol{o})}{\beta_m}A_m^i(o^i, a^i) + \sum_{k=1}^m\Delta_m^{k,*}(s)\log\tilde{\pi}_k^{i,*}(a^i|\tau^i, z_k^i). \tag{37}$$

Similar to the previous derivation, we aim to minimize the KL divergence:

$$\begin{aligned}
D_{KL}(\pi_m^{i,*}(\cdot|\tau^i, z_m^i)\|\pi_\theta(\cdot|\tau^i, z_m^i)) &= \sum_{a^i}\pi_m^{i,*}(a^i|\tau^i, z_m^i)\log\frac{\pi_m^{i,*}(a^i|\tau^i, z_m^i)}{\pi_\theta(a^i|\tau^i, z_m^i)} \\
&= -\sum_{a^i}\pi_m^{i,*}(a^i|\tau^i, z_m^i)\log\pi_\theta(a^i|\tau^i, z_m^i) + \text{const.}
\end{aligned} \tag{38}$$

We further estimate $z_k^i$ locally with $p_k$, and apply importance sampling using the behavior policy $\mu_m^i$ to transform the expectation over actions into an expectation over the offline dataset $\mathcal{D}_m$:

$$\begin{aligned}
&\mathbb{E}_{(\tau^i, a^i)\sim\mathcal{D}_m, z_m^i\sim q_m}\left[D_{KL}(\pi_m^{i,*}(\cdot|\tau^i, z_m^i)\|\pi_\theta(\cdot|\tau^i, z_m^i))\right] \\
&\propto -\mathbb{E}_{(s,\tau^i, a^i)\sim\mathcal{D}_m, z_m^i\sim q_m, \{\tilde{z}_k^i\}_{k=1}^m\sim p_m}\left[\exp\left(\tilde{A}_m^i(s, \boldsymbol{o}, a^i, \{\tilde{z}_k\}_{k=1}^m)\right)\cdot\log\pi_\theta(a^i|\tau^i, z_m^i)\right] + \text{const.} \\
&= L_{\pi,q}^{(2)}(\theta) + \text{const}
\end{aligned} \tag{39}$$

## A.3. Discussion

In this section, we discuss the implications of Theorem 4.1 and the key principles of COMAD for offline multi-agent continual skill discovery. Firstly, the optimal continual policy presents a geometric mixture of single task optimal policies with Lagrange multipliers being the *gating mechanism* for selecting useful skills. On the one hand, the scale of Lagrange multipliers determines the relative importance of different performance constraints: if for some $k$, $\lambda_k \gg \lambda_l, \forall l \neq k$, then the continual policy gives priority to the performance constraint of task $k$, approximately recovering the optimal single task policy $\tilde{\pi}_k$. On the other hand, the multipliers filter out useless skills by recognizing constraint conflicts, e.g., optimizing the policy for the performance on task 1 will degrade its performance on task 2. If the skills are not properly partitioned, the continual policy could suffer from mode collapse as shown in the toy example (Figure 2) and ablation studies (Figure 5).

Furthermore, the Lagrange multipliers are integrated into the reusability scores $\Delta_m^{k,*}(s)$, which provide fine-grained reusability information based on both performance constraint conflicts and confidence of states. Intuitively, the continual policy is expected to perform well on tasks that are similar (more reusable skills and less conflicts) and on states that are familiar (more confident decision-making) as depicted in the right panel of Figure 1(c). However, solving the complementary slackness condition for those multipliers requires accurate estimation of the expected returns $J_k(\pi_m)$ and the KL divergences $D_{KL}(\pi_m \| \mu_k)$ on all tasks, which is challenging in the offline task stream scenario where extrapolation error from multiple offline multi-agent datasets introduces vast bias and variance into the estimation. Hence the multi-head architecture is used as a simple yet efficient solution for approximating the multipliers to partition and reuse skills as depicted in our implementation (Figure 1(b)).

Lastly and critically, Theorem 4.1 presents a unified perspective for comparing and reusing skills across heterogeneous tasks through the geometric mixture of action distributions. Essentially, this mixture structure comes from the property of the KL divergence as a special case of the *Bregman Divergence*, which is defined for a strictly convex generator function $\phi$ as $D_\phi(p\|q) = \phi(p) - \phi(q) - \langle \nabla\phi(q), p - q \rangle$. Consequently, the minimization of a mixture of Bregman divergences, formulated as $\min_p \sum_k w_k D_\phi(p\|q_k), \sum_k w_k = 1$, is a convex optimization problem. Its first-order optimality condition corresponds to the solution in the dual space where the optimal solution is a linear combination of the reference points:

$$\nabla\phi(p^*) = \sum_k w_k \nabla\phi(q_k). \tag{40}$$

In the case of KL divergence, the generator is the negative entropy $\phi(p) = \sum p \log p$ with the gradient mapping $\nabla\phi(p) = 1 + \log p$. Consequently, the linear combination in the dual space yields $\log p^* = \sum w_k \log q_k$, which corresponds to the geometric mixture in the primal space.

# B. Experiment Details

## B.1. Environments and Designed Task Streams

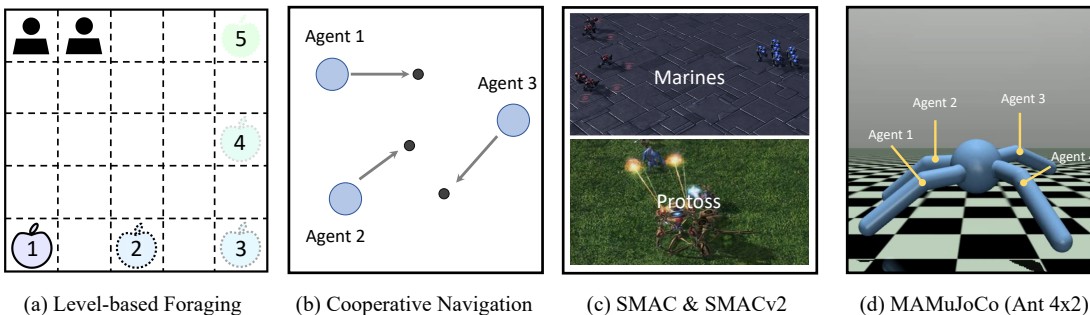

(a) Level-based Foraging      (b) Cooperative Navigation      (c) SMAC & SMACv2      (d) MAMuJoCo (Ant 4x2)

*Figure 6.* An illustration of the five environments used in this paper.

**Level-based Foraging** LBF is a classic cooperative MARL environment that emphasizes coordination. In this environment, multiple agents are placed in a 2D grid world with the objective of collecting food items scattered across the map. The core mechanism revolves around the level constraints: both agents and food items are assigned specific levels. A food item can only be successfully collected if the sum of the levels of all agents involved in the collection is greater than or

equal to the level of the food. This necessitates effective cooperation, as high-level food items require multiple agents to coordinate their arrival at the same location. To test the continual learning ability of the algorithms, we design a task stream based on the 2p1f scenario, where 2 agents start from random positions of the grid world and the only food is placed at five different locations: BottomLeft, Bottom, BottomRight, Right, TopRight as shown in Figure 6(a). This task stream features the property of changing objective, requiring the continual learning algorithm to adapt to different objectives flexibly.

**Cooperative Navigation** CN is a representative task within the Multi-Agent Particle Environment (MPE) (Mordatch & Abbeel, 2018), designed to test spatial coordination and collision avoidance in discrete spaces. In this task, $n$ agents are required to reach $n$ specific landmarks (Figure 6(b)). The primary challenge lies in the lack of predefined target assignment: agents must learn to autonomously decide which agent goes to which landmark to minimize the total distance of the group to all targets. Simultaneously, agents will be stopped for colliding with one another, requiring them to balance group safety with path efficiency. We design a stream of four tasks by changing the number of agents and landmarks respectively: CN-2, CN-3, CN-4, CN-5. This task stream specifically varies in the number of agents and landmarks, while the objective remains the same: reaching all landmarks, requiring the continual learning algorithm to fit agent team of different sizes.

**StarCraft Multi-Agent Challenge** SMAC is a micromanagement benchmark environment built upon the StarCraft II game engine. It is one of the most widely used platforms in the MARL community. In SMAC, each agent controls an individual unit with the goal of defeating an enemy team controlled by the built-in heuristic AI (Figure 6(c)). The environment offers diverse scenarios, including homogeneous matchups (e.g., 3m vs. 3m), heterogeneous matchups (e.g., 2s3z), and extreme asymmetric scenarios. Agents must learn sophisticated micromanagement tactics such as focus fire, kiting, and strategic positioning tailored to specific unit types. Due to its high-dimensional state space and the requirement for complex, coordinated decision-making, SMAC provides a challenging benchmark for MARL algorithms. To evaluate the continual learning algorithms on this benchmark, we design two types of task stream including homogeneous units (marines) and heterogeneous units (stalker and zealot), each contains 8 tasks from simple, symmetric to complex, asymmetric:

- **Marines** This stream contains a sequence of marine tasks: 3m, 4m, 10m, 12m, 5m_vs_6m, 7m_vs_8m, 9m_vs_10m, 13m_vs_15m;

- **Stalker-Zealot** This stream contains a sequence of stalker+zealot tasks: 1s1z, 2s3z, 2s4z, 3s5z, 4s4z, 2s1z_vs_3z, 2s2z_vs_4z, 4s4z_vs_8z.

**SMACv2** SMACv2 is an advanced version of the original SMAC designed to address the overfitting issues observed in the benchmark by introducing stochasticity. The original SMAC scenarios were found to be too static (e.g., fixed initial positions and unit compositions), leading algorithms to memorize specific trajectories rather than learning generalized combat intelligence. To combat this, SMACv2 introduces procedural generation, featuring randomized starting positions, randomized unit compositions, and more challenging observation masks. These additions demand high generalization capabilities, forcing algorithms to learn robust cooperative logic. To construct task streams, we change the number of allies and enemies and create 6 tasks for each of the three scenarios:

- **Protoss** 5v5, 6v6, 10v10, 20v20, 10v11, 20v23;

- **Terran** 5v5, 6v6, 10v10, 20v20, 10v11, 20v23;

- **Zerg** 5v5, 6v6, 10v10, 20v20, 10v11, 20v23.

**Multi-agent MuJoCo** MAMuJoCo is an extension of the classic single-agent MuJoCo (Todorov et al., 2012) continuous control tasks. The core idea is to decompose a complex robot (such as the Ant, HalfCheetah, or Humanoid) into multiple segments, where each segment is controlled by a different agent. Unlike the loose coordination found in other environments, MAMuJoCo features tightly coupled cooperation. Because all agents jointly control the same physical entity, a single erroneous action by one agent can cause the entire robot to lose balance or fall. This environment is characterized by continuous action spaces and highly non-linear dynamics, making it an essential tool for researching continuous-control MARL, hierarchical reinforcement learning, and fine-grained action alignment. Different from all above environments, MAMuJoCo provides an unbounded reward, thus we normalize the returns using the transformation: $R \leftarrow (R - R_{\min})/(R_{\max} - R_{\min})$, where we set $R_{\max} = 5000$ and $R_{\min} = 0$ based on the range of returns in the datasets (Table 6). We design two types of task streams based on the 4-agent Ant robot (Ant 4x2) depicted in Figure 6(d), whose ankles are controlled by 4 agents. The two streams concern varying objectives and varying dynamics:

*Table 2.* Details of datasets of 5 LBF tasks.

| Task | Bottom | | BottomLeft | | BottomRight | | Right | | TopRight | |
|---|---|---|---|---|---|---|---|---|---|---|
| Quality | Expert | Medium | Expert | Medium | Expert | Medium | Expert | Medium | Expert | Medium |
| Average Length | 39.11 | 38.48 | 5.66 | 42.54 | 10.44 | 24.74 | 6.86 | 13.92 | 5.65 | 18.49 |
| Average Return | 0.95 | 0.34 | 1.00 | 0.51 | 1.00 | 0.68 | 1.00 | 0.94 | 1.00 | 0.78 |

*Table 3.* Details of datasets of 4 CN tasks.

| Task | CN-2 | | CN-3 | | CN-4 | | CN-5 | |
|---|---|---|---|---|---|---|---|---|
| Quality | Expert | Medium | Expert | Medium | Expert | Medium | Expert | Medium |
| Average Length | 3.11 | 36.36 | 14.16 | 33.88 | 18.21 | 36.91 | 18.33 | 33.35 |
| Average Return | 1.00 | 0.47 | 0.95 | 0.50 | 0.88 | 0.39 | 0.40 | 0.25 |

- **Reward** This task stream contains 8 tasks, which vary in the reward functions of tasks, requiring the agent to move forward as fast as possible in the degrees of 0 (the original objective), 45, 90, 135, 180, 225, 270, and 315, respectively.

- **Dynamics** This task stream contains 6 non-monotonic tasks, which are implemented by disabling controllable legs of the ant agent: Back, Diagonal (Back + Front), Front, Mixed (all except the front leg), Right, Normal. The reward function is fixed, requiring the agent to move in the direction of 0 degree as fast as possible.

Note that here we arrange all tasks generally from easier to harder for a unified evaluation protocol. Yet task ordering has a noticeable effect on continual learning performance, as prior work suggests that orders maximizing task dissimilarity can be optimal (Li & Hiratani, 2025; Bell & Lawrence, 2022). In our setting, proper task ordering may help learn more reusable skills and we conduct ablation studies in Section D.3 for further analysis.

### B.2. Data Collection and Datasets

We construct offline datasets for all except MaMuJoCo tasks in the designed task streams based on the PyMARL implementation[1] of the MARL algorithm QMIX (Rashid et al., 2018), and construct the offline datasets of MaMuJoCo tasks based on the HARL implementation[2] of the continuous MARL algorithm HATD3 (Zhong et al., 2024). Considering the large scale of the datasets, we only collect data with two types of qualities: expert and medium. Their properties are listed below:

- The **Expert** dataset contains trajectory data collected by QMIX or HATD3 policies trained by 2M steps and 10M steps respectively. The statistics info is recorded to aid constructing the medium datasets.

- The **Medium** dataset contains trajectory data collected by QMIX or HATD3 policies, whose training processes are stopped when the evaluated win rates (SMAC) or returns (other environments except SMAC and SMACv2) exceed half of the corresponding expert policies. Note that for SMACv2, QMIX cannot reach very high win rates as in SMAC tasks, hence we only collect medium datasets with the converged policy trained by QMIX.

We also record the details of these datasets including the average trajectory length and the average returns, as shown in Table 2, 3, 4, 5 and 6. Note that all dataset are processed to have exactly 2000 trajectories each.

---

[1] https://github.com/oxwhirl/pymarl
[2] https://github.com/PKU-MARL/HARL

*Table 4.* Details of datasets of 16 SMAC tasks.

| Task | 3m | | 4m | | 10m | | 12m | | 5m_vs_6m | | 7m_vs_8m | | 9m_vs_10m | | 13m_vs_15m | |
|---|---|---|---|---|---|---|---|---|---|---|---|---|---|---|---|---|
| Quality | Expert | Medium | Expert | Medium | Expert | Medium | Expert | Medium | Expert | Medium | Expert | Medium | Expert | Medium | Expert | Medium |
| Average Length | 23.86 | 23.89 | 26.68 | 26.73 | 30.68 | 31.16 | 27.66 | 32.42 | 28.64 | 30.97 | 27.70 | 28.41 | 31.79 | 30.46 | 32.68 | 31.34 |
| Average Return | 19.96 | 14.83 | 19.74 | 13.37 | 19.94 | 16.13 | 19.95 | 17.92 | 17.67 | 14.37 | 19.06 | 15.72 | 19.66 | 15.50 | 19.77 | 16.53 |
| Average Win Rate | 1.00 | 0.54 | 1.00 | 0.42 | 1.00 | 0.51 | 1.00 | 0.69 | 0.66 | 0.44 | 0.88 | 0.46 | 0.94 | 0.42 | 0.94 | 0.43 |

| Task | 1s1z | | 2s3z | | 2s4z | | 3s5z | | 4s4z | | 2s1z_vs_3z | | 2s2z_vs_4z | | 4s4z_vs_8z | |
|---|---|---|---|---|---|---|---|---|---|---|---|---|---|---|---|---|
| Quality | Expert | Medium | Expert | Medium | Expert | Medium | Expert | Medium | Expert | Medium | Expert | Medium | Expert | Medium | Expert | Medium |
| Average Length | 41.48 | 36.34 | 47.48 | 57.92 | 48.81 | 57.64 | 56.79 | 62.51 | 56.18 | 61.08 | 53.31 | 51.07 | 53.25 | 46.00 | 73.18 | 88.47 |
| Average Return | 19.72 | 14.97 | 19.72 | 17.69 | 19.87 | 17.15 | 19.81 | 16.82 | 19.79 | 16.60 | 20.02 | 15.93 | 19.66 | 16.59 | 19.30 | 17.05 |
| Average Win Rate | 0.94 | 0.44 | 0.95 | 0.65 | 0.98 | 0.55 | 0.96 | 0.47 | 0.91 | 0.41 | 0.97 | 0.47 | 0.94 | 0.63 | 0.84 | 0.41 |

*Table 5.* Details of datasets of 18 SMACv2 tasks.

| Task | Protoss-5v5 | Protoss-6v6 | Protoss-10v10 | Protoss-20v20 | Protoss-10v11 | Protoss-20v23 |
|---|---|---|---|---|---|---|
| Quality | Medium | | | | | |
| Average Length | 68.66 | 69.19 | 76.33 | 87.85 | 80.58 | 81.02 |
| Average Return | 15.95 | 17.09 | 19.65 | 14.88 | 14.97 | 12.56 |
| Average Win Rate | 0.66 | 0.72 | 0.57 | 0.31 | 0.31 | 0.11 |

| Task | Zerg-5v5 | Zerg-6v6 | Zerg-10v10 | Zerg-20v20 | Zerg-10v11 | Zerg-20v23 |
|---|---|---|---|---|---|---|
| Quality | Medium | | | | | |
| Average Length | 31.64 | 34.20 | 37.48 | 42.99 | 36.72 | 42.61 |
| Average Return | 14.93 | 12.76 | 14.83 | 16.54 | 11.82 | 11.84 |
| Average Win Rate | 0.59 | 0.34 | 0.57 | 0.56 | 0.22 | 0.19 |

| Task | Terran-5v5 | Terran-6v6 | Terran-10v10 | Terran-20v20 | Terran-10v11 | Terran-20v23 |
|---|---|---|---|---|---|---|
| Quality | Medium | | | | | |
| Average Length | 52.86 | 60.87 | 70.32 | 80.74 | 62.61 | 69.94 |
| Average Return | 13.83 | 17.15 | 18.02 | 16.89 | 13.54 | 15.07 |
| Average Win Rate | 0.71 | 0.72 | 0.71 | 0.60 | 0.38 | 0.34 |

*Table 6.* Details of datasets of 14 MaMuJoCo tasks.

| Task | Reward-0 | | Reward-45 | | Reward-90 | | Reward-135 | | Reward-180 | | Reward-225 | | Reward-270 | |
|---|---|---|---|---|---|---|---|---|---|---|---|---|---|---|
| Quality | Expert | Medium | Expert | Medium | Expert | Medium | Expert | Medium | Expert | Medium | Expert | Medium | Expert | Medium |
| Average Length | 999.66 | 999.95 | 999.95 | 999.36 | 999.95 | 999.90 | 999.72 | 999.23 | 999.89 | 999.33 | 999.80 | 1000.00 | 999.69 | 999.57 |
| Average Return | 4906.74 | 3553.48 | 4792.56 | 2711.80 | 4650.29 | 3623.93 | 4815.57 | 3317.08 | 5177.54 | 3834.87 | 5241.61 | 3047.27 | 5130.75 | 3453.15 |

| Task | Reward-315 | | Dynamics-Back | | Dynamics-Diagonal | | Dynamics-Front | | Dynamics-Mixed | | Dynamics-Right | | Dynamics-Normal | |
|---|---|---|---|---|---|---|---|---|---|---|---|---|---|---|
| Quality | Expert | Medium | Expert | Medium | Expert | Medium | Expert | Medium | Expert | Medium | Expert | Medium | Expert | Medium |
| Average Length | 999.85 | 999.05 | 999.74 | 999.48 | 999.70 | 999.73 | 999.65 | 999.18 | 999.97 | 1000.02 | 1000.03 | 999.79 | 999.82 | 999.01 |
| Average Return | 5161.54 | 3659.56 | 3048.77 | 2711.17 | 4061.96 | 2972.82 | 5118.01 | 2638.49 | 2182.55 | 1058.68 | 3707.13 | 2754.11 | 5600.11 | 4004.47 |

### B.3. Baselines

**Continual Offline MARL Baselines**    As there is no dedicated existing continual offline MARL algorithm, we adapt 6 representative continual learning baselines and continual reinforcement learning algorithms based on the framework of the OMIGA (Wang et al., 2023) method. Concretely, OMIGA framework contains critic and actor. We always re-initialize the critic (including the Q, V function and the mixer module) by adding a noise $\epsilon$ of controlled norm $\|\epsilon\|_2 \leq 0.01$ to improve plasticity, inspired by the observation that the learning of critics may be sensitive to biases (Wolczyk et al., 2022). As for the actor, we take different strategies and adapt the following algorithms:

- **Fine-tune (FT)** No extra strategy is applied to the actor, which is optimized sequentially for each task in the task stream. FT is treated as a soft lower bound.

- **From Scratch (FS)** The actor is re-initialized when switch to a new task. FS can be regarded as the upper bound for plasticity, but no transfer exists in the actor level.

- **Multi Task (MT)** The actor and the critic are trained with randomly sampled multi-task data at the same time, representing a soft upper bound.

- **Elastic Weight Consolidation (EWC)** A regularization-based approach that penalizes changes to the parameters of the actor based on its importance for old tasks. The importance of weights is estimated via the Fisher Information Matrix, balancing the preservation of old knowledge with the plasticity for new tasks.

- **Continual RL without Conflict (OWL)** A method that implements the actor based on a multi-head architecture and a shared feature extractor to minimize interference among different tasks while transfer its learned representations via the shared feature extractor, which is regularized based on the changes to its parameters. Note that OWL always creates an output head for each task and has no explicit transfer among tasks. While our method allocate output heads based on the estimated state density and transfer among tasks via a skill-augmented objective.

- **Rehearsal** An experience replay-based strategy that maintains a buffer of trajectories from prior tasks. During the training of a new task, samples from this buffer are periodically replayed to the actor and critic to prevent catastrophic forgetting of past experiences. In our implementation, we always keep 128 trajectories of old tasks and train on these experiences every two training steps.

**Multi-agent Skill discovery Baselines**    As far as we know, currently there is no skill discovery algorithm for continual offline MARL, so we introduce the following two state-of-the art baselines from offline multitask MARL literature:

- **ODIS** The algorithms proposes a VAE structure to extract informative coordination skill from multiple offline multi-agent datasets simultaneously, and then leverage a hierarchical coordination policy to choose optimal coordination skills, effectively enhancing generalization on unseen tasks.

- **HiSSD** The algorithm leverages a hierarchical framework to jointly extract common coordination skills and task-specific skills to facilitate generalization on unseen tasks.

Note that both ODIS and HiSSD construct a fixed-size skill library through pretraining on multi-agent offline datasets, and rely on the generalization ability of this skill library to adapt to downstream tasks in a zero-shot manner. To align with the continual learning setting, we pretrain ODIS on the first tasks of each task stream, and pretrain HiSSD on the first two tasks of each task stream due to its use of contrastive loss between tasks.

## C. Implementation Details

### C.1. Network Architecture and Hyperparameters

The network architecture of the proposed COMAD framework follows a structural design pattern:

- **Input Processing** To process the local observations $o^i$, actions $a^i$ and global state $s$ with varying length, we employ a self-attention mechanism for all critic and actor modules by first projecting inputs into a latent space via linear embedding layers, and then applying the self-attention shown in Equation (4) to obtain fixed-length embeddings that capture inter-agent relationships.

---

**Algorithm 2** COMAD Training

---

 1: **Input:** Offline Datasets $\mathcal{D} = \{D_1, \ldots, D_M\}$ (Sequentially)
 2: **Initialize:** Feature extractor $F$, critics $Q, V$, mixer $w, b$, target networks, density estimator feature extractor $F^E$.
 3: **for** task $m = 1$ to $M$ **do**
 4:     **// Head Expansion**
 5:     **if** $m = 1$ **then**
 6:         Allocate heads $G_1^\pi, G_1^p, G_1^E$.
 7:         Set active head index $k^* \leftarrow 1$.
 8:     **else**
 9:         Evaluate expected state density $\hat{d}_k \approx \mathbb{E}_{s \sim D_m}[d_k(s)]$ for old heads $k < m$.
10:         **if** $\exists k : \hat{d}_k > d_0$ **then**
11:             Reuse old heads: $k^* \leftarrow \arg\max_k \mathbb{E}[d_k(s)]$.
12:         **else**
13:             Expand new heads $G_m^\pi, G_m^p, G_m^E$.
14:             Set active head index $k^* \leftarrow m$.
15:         **end if**
16:     **end if**
17:     **// Optimization Loop**
18:     **for** sampled batch $B \sim D_m$ **do**
19:         Update $Q, V, w, b$ by minimizing critic losses in Equation (2) and (3).
20:         Update local encoder $p_{k^*}$ by minimizing KL loss in Equation (6).
21:         Calculate NCE loss $L_{NCE}$ via Equation (10).
22:         **if** $m = 1$ or in Stage 1 **then**
23:             $L_{\pi,q} \leftarrow$ vanilla policy loss in Equation (5).
24:         **else**
25:             $L_{\pi,q} \leftarrow$ skill-augmented policy loss in Equation (11).
26:         **end if**
27:         **if** $m \geq 2$ **then**
28:             Apply regularization: $L_{\pi,q} \leftarrow L_{\pi,q} + \lambda_{reg}\|\theta_{F,m} - \theta_{F,1}\|_2^2$, $L_{NCE} \leftarrow L_{NCE} + \lambda_{reg}\|\theta_{F^E,m} - \theta_{F^E,1}\|_2^2$.
29:         **end if**
30:         Update state density estimator $E_{k^*}$ by minimizing $L_{NCE}$.
31:         Update action decoder $\pi_{k^*}$ and global encoder $q_{k^*}$ by minimizing $L_{\pi,q}$.
32:     **end for**
33:     **if** $m = 1$ **then**
34:         Save checkpoints $\theta_{F,1}$ and $\theta_{F^E,1}$.
35:     **end if**
36: **end for**

---

 

---

**Algorithm 3** COMAD Evaluation

---

 1: **Input:** Target environment and state samples (states $S_{eval}$)
 2: **for** each available head $k$ **do**
 3:     Evaluate expected state density $\hat{d}_k \approx \mathbb{E}_{s \sim S_{eval}}[d_k(s)]$ using $E_k$.
 4: **end for**
 5: Select execution head: $k^* \leftarrow \arg\max_k \hat{d}_k$.
 6: **while** episode not done **do**
 7:     Observe local observation $o^i$ and history $\tau^i$ for each agent $i$.
 8:     Generate skill embedding: $z^i \sim p_{k^*}(z^i|\tau^i)$.
 9:     Generate action: $a^i \sim \pi_{k^*}^i(a^i|\tau^i, z^i)$.
10:     Execute joint action $\boldsymbol{a} = (a^1, \cdots, a^n)$ in environment.
11: **end while**

---

*Table 7.* Hyperparameters of COMAD for all experiments.

| Hyperparameter | Value |
|---|---|
| KL regularization weight $\alpha$ | 10 |
| Attention projection dimension | 8 |
| MLP hidden dimension | 64 |
| MLP depth | 3 |
| GRU hidden dimension | 64 |
| GRU depth | 1 |
| Skill dimension $\dim(z_m^i)$ | 16 |
| $L_2$ regularization strength $\lambda_{reg}$ | 500 |
| NCE noise scale $\sigma_s$ | 0.1 |
| Confidence threshold $d_0$ | 2 (CN) |
| | 8 (LBF, SMAC, SMACv2, MaMuJoCo) |
| Number of tasks $M$ | $4 \sim 8$ |
| Training steps per task $\Delta$ | $2 \times 10^4$ (LBF, CN, SMAC, SMACv2) |
| | $1 \times 10^5$ (MaMuJoCo) |
| Optimizer | AdamW |
| Learning rate | $5 \times 10^{-4}$ |
| Weight decay | $1 \times 10^{-3}$ |
| Batchsize | 32 |
| Target network update ratio | 0.005 |

- **Sequence Modeling** We utilize a single layer Gated Recurrent Unit (GRU) (Chung et al., 2014) to model the trajectories for the actor. Concretely, the embeddings extracted by the self-attention mechanism are fed into an MLP, followed by the GRU module to output the sequence embeddings $\tau^i$, which is used to output actions via the output heads; and $h^i$, which is used by the GRU to transmit sequential information.

- **Single Step Modeling** As for the critics including the Q, V, mixer networks and their target networks, the inputs correspond to single steps (such as $o^i$, $s$). These networks directly feed the embeddings obtained via the self-attention mechanism to MLPs and output the values (Q, V values or the weight and bias of the mixing networks).

- **Multi-Head Modules** As presented in Figure 1(b) of the main text, the skill prior, policy and the confidence module are equipped with multi-head modules of the same structure. Concretely, each head in these modules is a three-layer MLP with hidden dim 64 (approximately 10K parameters) that outputs skill priors $z_m^i$, action distributions $\pi_m^i$ and confidence scores $\Delta_m^k$ respectively. Moreover, the multi-head module support operations of head allocation and switch, providing a consistent and simple interface for the algorithm.

We summarize our setting of hyperparameters in Table 7. Note that for our method COMAD, training steps of both stage 1 and stage 2 are half of the total training steps per task $\Delta$.

### C.2. Training and Evaluation

In continual learning problem, the training and evaluation process proceed alternately as the datasets of the task stream are provided in a sequential manner. Here we split the training and evaluation process of COMAD into two phases as shown in Algorithm 2 and 3 respectively. In the training phase, COMAD first identifies if new output heads are needed (Line $5 \sim 16$). Then the selected head $k^*$ is used in the optimization loop for the current task (Line $18 \sim 30$). To preserve the memory of feature extractors $F, F^E$, the algorithm saves the checkpoints of their parameters $\theta_{F,1}, \theta_{F^E,1}$ at the end of the training of the first task. In the evaluation phase, COMAD first identifies the best output head via a small sample of states from the target environment, then executes in a decentralized manner with local encoder $p_{k^*}(z^i|\tau^i)$ and action decoder $\pi_{k^*}^i(a^i|\tau^i, z^i)$.

## D. Additional Results

In this section we present the full results on all environments, including additional ablation studies on Marines-expert task of SMAC (Section D.1), the sensitivity studies of two critical hyperparameters $\sigma_s$ and $d_0$ (Section D.2), the effect of task ordering (Section D.3), the growth of the skill library (Section D.4), the forward transfer and backward transfer metrics (Section D.5), and the learning curves of average performance P($t$) of all methods (Section D.6).

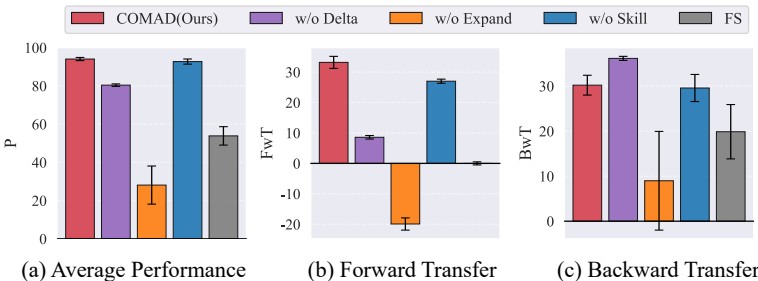

*Figure 7.* Metrics of ablation studies on Marines-expert task.

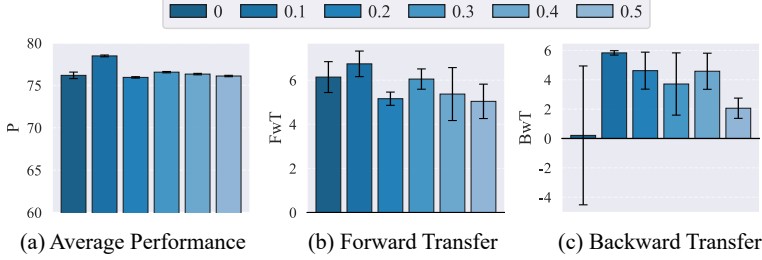

*Figure 8.* Sensitivity analysis of the noise scale $\sigma_s$ on CN-expert task.

## D.1. Additional Ablation Studies on Marines

The results of ablation studies on Marines-expert task are presented in Figure 7. In contrast to CN tasks, SMAC tasks emphasize task heterogeneity and thus a fixed-size skill library (w/o Expand) fails to accommodate the evolving coordination skills required for new team configuration, leading to transferability loss and poor overall performance. On the other hand, removing the skill encoders (w/o Skill) results in only minor degradation. We hypothesize that this contributes to the simplicity of single task dataset, on which a monolithic policy is sufficient for learning performant skills. Importantly, the absence of reusability estimation (w/o Delta) significantly impairs forward transfer due to indiscriminative skill reuse and the resulting interference. Despite this, it slightly improves backward transfer, but we interpret this as a misleading signal. Since w/o Delta suffers from poor initial performance for each task, it leaves a larger margin for backward improvement via the shared feature extractor during the training of subsequent tasks. Thus the high backward transfer of w/o Delta reflects the recovery from a sub-optimal starting point rather than superior knowledge retention.

## D.2. Sensitivity Studies

We conduct sensitivity studies on CN-expert task to investigate the influence of the two critical hyperparameters within COMAD: the NCE noise scale $\sigma_s$ and the confidence threshold $d_0$. The results are shown in Figure 8 and Figure 9, showing that the default choices for CN-expert task ($\sigma_s = 0.1, d_0 = 2$) achieves the best performance. When setting a higher noise scale, the estimator tends to focus on noises and gradually lose its capability to distinguish the difference of states among tasks. Besides, when $\sigma_s = 0$, the estimator is unable to identify tasks, leading to the lowest BwT. As for the confidence threshold, lower $d_0$ indicates skill reuse without filtering and higher $d_0$ means more strict reuse, both of which lead to downgraded P and BwT. However, due to the similarity among CN tasks, FwT metric is not sensitive to $d_0$.

## D.3. Task Ordering

We conduct experiments on CN-expert and Marines-expert to investigate the effect of task ordering. In the following, Seq 0 stands for the original difficulty order, Seq 1 reverses Seq 0 (from harder to easier), Seq 2 and Seq 3 randomly scatter the task sequences:

- **CN Seq 2**: CN-3, CN-5, CN-2, CN-4;

- **CN Seq 3**: CN-4, CN-2, CN-5, CN-3;

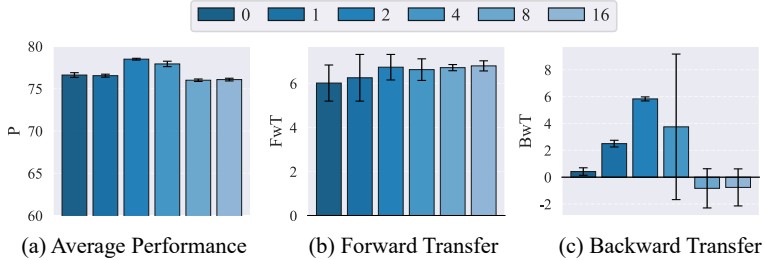

*Figure 9.* Sensitivity analysis of the confidence threshold $d_0$ on CN-expert task.

*Table 8.* Metrics $\pm$ std of different algorithms on different task ordering of CN-expert and Marines-expert.

| CN-expert | Seq 0 | Seq 1 | Seq 2 | Seq 3 | Marines-expert | Seq 0 | Seq 1 | Seq 2 | Seq 3 |
|---|---|---|---|---|---|---|---|---|---|
| P | 78.49±0.41 | 84.27±0.49 | 82.81±2.81 | 88.13±1.25 | P | 94.05±0.83 | 95.66±4.96 | 93.36±10.68 | 96.75±4.61 |
| FwT | 6.74±0.90 | 7.95±0.71 | 7.83±0.45 | 8.38±0.22 | FwT | 33.20±1.98 | 35.52±3.64 | 36.81±2.25 | 31.67±4.05 |
| BwT | 5.83±0.83 | 6.27±0.31 | 6.05±2.92 | 5.96±3.75 | BwT | 30.80±2.20 | 27.52±5.93 | 13.43±2.68 | 24.72±5.16 |

- **Marines Seq 2**: 12m, 3m, 7m_vs_8m, 13m_vs_15m, 4m, 9m_vs_10m, 10m, 5m_vs_6m;

- **Marines Seq 3**: 5m_vs_6m, 10m, 13m_vs_15m, 4m, 9m_vs_10m, 3m, 12m, 7m_vs_8m.

The results in Table 8 suggest that starting from medium-scale tasks (e.g., Seq 3 in both CN and SMAC) appears more beneficial than the difficulty order, indicating that these tasks may provide more reusable skills for later transfer. Nonetheless, COMAD remains effective for these four sequences, showcasing its robustness to the task ordering.

### D.4. Growth of the Skill Library

To illustrate the growth of the skill library, we extend our experiment to repeated-CN-expert and Marines-expert and present the number of output heads, the maximum reusability scores and the reused heads in Table 9 and Table 10. We observe that in CN($d_0 = 2$) only 2 heads are allocated and 3 reuses occur in Marines($d_0 = 8$), indicating that the number of heads grows controllably by setting the confidence threshold $d_0$. With higher $d_0$, the expansion will be sparser and the growth can be effectively limited.

### D.5. Forward and Backward Transfer

We list here the results of forward and backward transfer in Table 11 and Table 12 respectively, which are complementary to the main results in Table 1. We also report Fwt $\times$ 100 and BwT $\times$ 100 for numerical consistency. Note that MT is omitted as it learns all task simultaneously and FS is omitted in FwT results, which is used as baseline (i.e., FwT of FS is always zero).

The FwT results showcase similar trend compared to the results of average performance: skill discovery methods (ODIS and HiSSD) with fixed-size skill libraries are limited due to plasticity loss, suggesting that they cannot flexibly learn new skills and perform worse than FS. Besides, FT, EWC and Rehearsal also lose their plasticity due to overfitting (FT), parameter regularization (EWC) or gradient conflict (Rehearsal). Instead, OWL achieves positive forward transfer thanks to parameter isolation, while COMAD exceeds it by actively reusing skills.

The BwT results of skill discovery baselines (ODIS, HiSSD) are relatively higher due to the fixed skill set, while vanilla methods (FS, FT) fail to memorize previous skills and obtain the lowest backward transfer. Interestingly, EWC behaves differently, mitigating catastrophic forgetting on expert dataset of LBF, SMAC-Marines and MAMuJoCo-Reward compared to FS, while exacerbating forgetting on most medium datasets. We hypothesize that the observed degradation in BwT for EWC can be attributed to its sensitivity to dataset quality, thereby affecting the estimation of the Fisher information matrix. Rehearsal still faces the problem of gradient conflict and obtain mediocre backward transfer. OWL achieves positive backward transfer by implicitly utilizing the similarity among tasks while COMAD promotes backward transfer through skill partition and finetuning via the shared feature extractor.

*Table 9.* Average performances, the number of output heads, maximum reusability scores and the reused heads in the learning process of the repeated-CN-expert task stream.

| repeated-CN-expert | CN-2 | CN-3 | CN-4 | CN-5 | CN-2 | CN-3 | CN-4 | CN-5 |
|---|---|---|---|---|---|---|---|---|
| P | 24.82 | 48.13 | 64.76 | 78.32 | 76.47 | 77.41 | 79.71 | 78.95 |
| n_head | 1 | 2 | 2 | 2 | 2 | 2 | 2 | 2 |
| max reuse score | N/A | 1.73 | 2.57 | 2.35 | 2.62 | 2.58 | 2.47 | 2.39 |
| reused head | new | new | CN-3 | CN-3 | CN-2 | CN-3 | CN-3 | CN-3 |

*Table 10.* Average performances, the number of output heads, maximum reusability scores and the reused heads in the learning process of the Marines-expert task stream.

| Marines-expert | 3m | 4m | 10m | 12m | 5m_vs_6m | 7m_vs_8m | 9m_vs_10m | 13m_vs_15m |
|---|---|---|---|---|---|---|---|---|
| P | 12.5 | 24.35 | 36.77 | 49.08 | 56.67 | 69.32 | 81.64 | 94.05 |
| n_head | 1 | 1 | 2 | 2 | 3 | 3 | 4 | 5 |
| max reuse score | N/A | 8.56 | 5.83 | 9.34 | 4.62 | 8.07 | 6.47 | 5.53 |
| reused head | new | 3m | new | 10m | new | 5m_vs_6m | new | new |

## D.6. Full Learning Curves

We list here the curves of average performances on all tasks in Figure 10 (LBF and CN), Figure 11 (SMAC and SMACv2) and Figure 12 (MAMuJoCo). Note that FwT can be understood as the area between one curve and the curve of FS.

The results of learning curves present the same observations that skill discovery baselines fail to learn new skills, vanilla methods and EWC are relatively limited due to catastrophic forgetting or plasticity loss, OWL facilitates forward and backward transfer via parameter isolation while COMAD achieves the best overall bidirectional transfer via active skill partition and reuse.

Besides, we can also observe the influence of the two parts design (symmetric and asymmetric) in SMAC tasks, where the last four (Marines) or three (Stalker-Zealot) tasks of each task stream belong to the asymmetric type. Under these heterogeneous task streams, vanilla methods face the problem of balancing between symmetric task skills and asymmetric ones, where FT and Rehearsal become unstable as depicted in Marines tasks and Stalker-Zealot medium task.

## E. Limitations and Future Work

COMAD is proposed as a principled framework for continual offline multi-agent skill discovery under the task-bounded setting. Despite its strong empirical performance on the controlled experiments, it has several limitations and can be further improved for higher flexibility and broader applications.

First, the reusability score combines the Lagrange multipliers and state densities as a proxy for the ideal skill-gating mechanism by reusing skills based on task similarity and state familiarity. It captures whether the current task is similar to any previous task and whether the current state falls within the support of previously learned skills. However, tasks in open-ended environments may require reusing skills based on fine-grained semantic features such as human feedback or natural language guidance. Hence a human-in-the-loop continual reinforcement learning paradigm is an important direction for future work.

Second, COMAD focuses on the task-bounded setting, allowing us to study transfer, forgetting and skill library expansion in a controlled manner, but limits its applicability to settings with unknown or ambiguous task boundaries. Nonetheless, the reusability score can be potentially extended to the task-unbounded setting by implicitly modeling the task boundaries. For example, two tasks can be identified as similar if their performance constraints are positively correlated and thus can both be learned without knowing task boundaries; meanwhile task-agnostic coordination patterns could be extracted by disentangling the action distribution at the same state across different tasks. Based on the above task-agnostic components, we may actively allocate, prune or merge the output heads, maintaining a skill library with stable memory, plasticity and controlled size (Pan et al., 2025).

More broadly, evaluating whether the same skill partition-and-reuse principles continue to hold in more open-ended environments such as multi-agent embodied intelligence, larger models such as Vision-Language-Action (VLA) models and semantically complex environments such as human-AI coordination, remains an important next step.

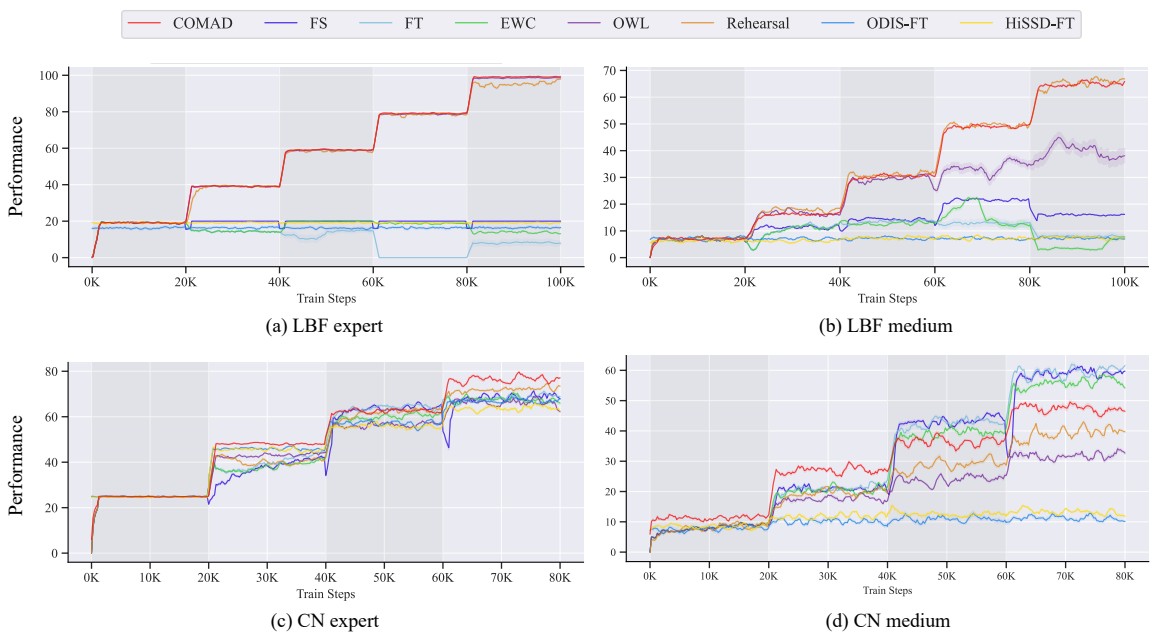

*Figure 10.* Learning curves on LBF and CN tasks.

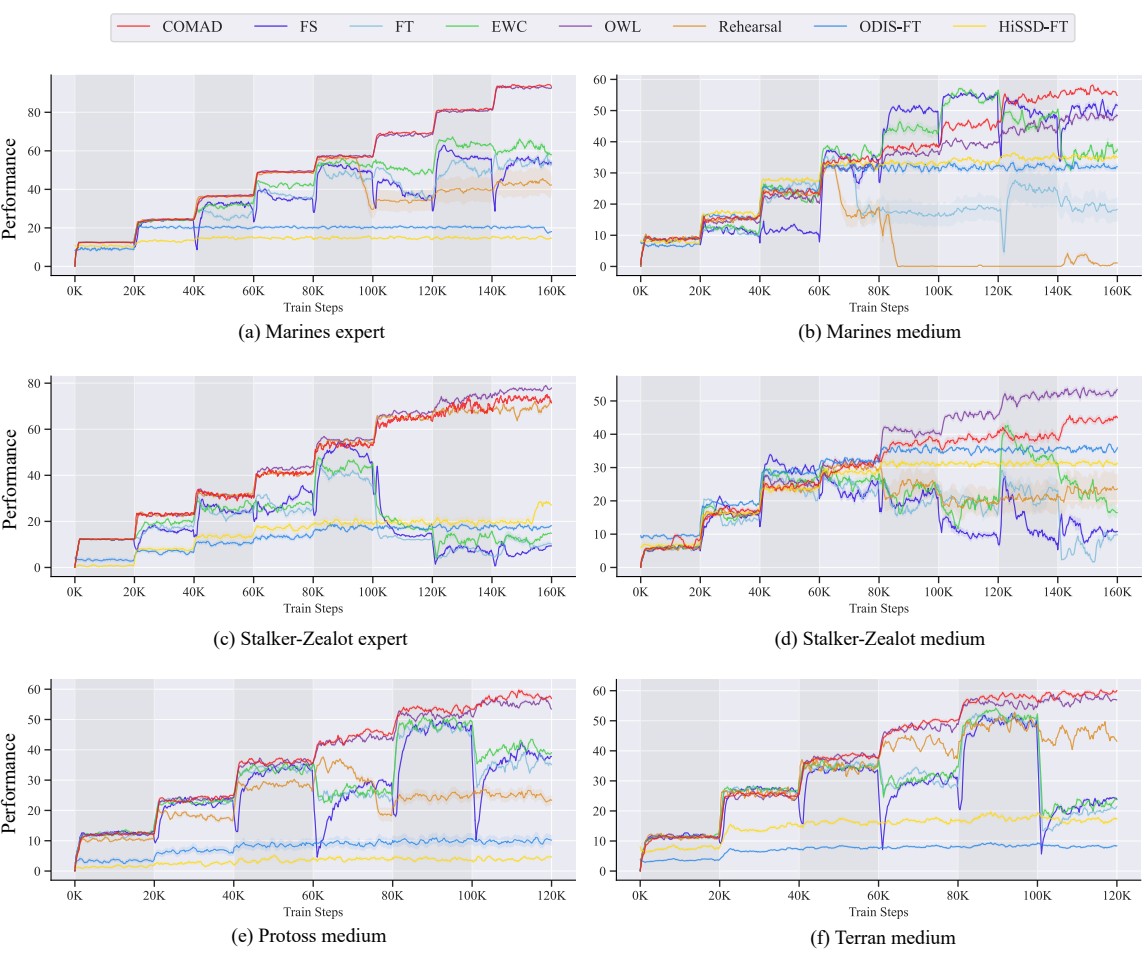

*Figure 11.* Learning curves on SMAC and SMACv2 tasks.

*Table 11.* Forward Transfer ± std of different algorithms on task streams from LBF, CN, SMAC, SMACv2 and MAMuJoCo environments. The best and second-best methods are marked in **bold** and underlined, respectively. An asterisk (*) indicates that a method's mean performance is no lower than the best mean minus one standard deviation of the best method in the same row. All results are based on 5 distinct seeds and 32 episodes per seed on each evaluation step. "Overall" reports the average forward transfer of each algorithm over all task streams.

| Task Stream | Dataset | COMAD(ours) | FT | EWC | OWL | Rehearsal | ODIS-FT | HiSSD-FT |
|---|---|---|---|---|---|---|---|---|
| LBF | Expert | **-0.42 ± 0.57** | -46.18 ± 18.77 | -12.57 ± 0.92 | -0.75 ± 0.11* | -3.47 ± 0.80 | -80.38 ± 2.88 | -77.55 ± 0.04 |
| | Medium | -3.80 ± 0.91 | -19.40 ± 16.00 | -22.41 ± 8.02 | -17.30 ± 13.47 | **-0.36 ± 0.53** | -57.78 ± 0.12 | -57.57 ± 1.87 |
| CN | Expert | **6.74 ± 0.90** | 1.29 ± 1.15 | -2.48 ± 3.80 | -4.33 ± 0.11 | -0.38 ± 0.30 | -2.74 ± 0.64 | -5.55 ± 2.29 |
| | Medium | **16.00 ± 0.39** | 0.71 ± 0.38 | -0.89 ± 0.14 | -0.82 ± 0.64 | -2.50 ± 0.28 | -19.34 ± 2.43 | -17.36 ± 1.21 |
| Marines | Expert | **33.20 ± 1.98** | -3.75 ± 0.27 | 5.57 ± 0.30 | 29.83 ± 0.27 | 13.73 ± 13.16 | -16.56 ± 2.69 | -22.08 ± 0.06 |
| | Medium | **10.97 ± 3.18** | -11.03 ± 9.26 | -0.64 ± 1.08 | 0.81 ± 10.94 | -18.21 ± 3.49 | 0.31 ± 3.66 | 2.90 ± 0.55 |
| Stalker-zealot | Expert | 24.50 ± 0.79 | -2.47 ± 1.39 | -1.47 ± 0.36 | **27.55 ± 0.68** | 0.74 ± 3.57 | -12.93 ± 1.70 | -10.73 ± 7.82 |
| | Medium | 16.64 ± 3.27 | 2.67 ± 0.06 | 2.89 ± 1.10 | **22.74 ± 1.12** | 20.28 ± 0.36 | 18.22 ± 0.61 | 11.73 ± 1.16 |
| Protoss | Medium | **13.79 ± 1.60** | -0.90 ± 0.09 | 2.07 ± 0.57 | 13.42 ± 0.72* | 0.23 ± 7.61 | -29.97 ± 0.42 | -23.96 ± 0.89 |
| Zerg | Medium | **12.76 ± 1.37** | 2.55 ± 0.60 | 3.15 ± 0.39 | 12.48 ± 1.10* | -3.25 ± 11.07 | -38.27 ± 3.18 | -37.48 ± 5.56 |
| Terran | Medium | **7.27 ± 1.35** | -3.66 ± 0.13 | -11.27 ± 6.10 | -0.24 ± 5.35 | -5.30 ± 3.66 | -30.22 ± 9.72 | 3.46 ± 3.18 |
| Reward | Expert | **36.48 ± 1.11** | -3.13 ± 0.41 | -0.50 ± 5.36 | 34.38 ± 0.27 | -7.59 ± 3.67 | -5.31 ± 1.28 | -5.53 ± 0.21 |
| | Medium | **27.15 ± 0.18** | 1.80 ± 0.11 | -0.65 ± 1.58 | 17.20 ± 4.45 | 0.69 ± 0.14 | 1.18 ± 0.54 | 0.73 ± 0.31 |
| Dynamics | Expert | 1.44 ± 2.35 | 3.82 ± 4.31* | -2.82 ± 0.23 | **4.17 ± 0.67** | -11.30 ± 0.24 | 0.12 ± 0.36 | 2.10 ± 0.49 |
| | Medium | -3.80 ± 1.47 | 0.57 ± 0.42* | 0.90 ± 0.66* | -4.02 ± 1.82 | 0.98 ± 0.03* | 0.63 ± 0.25* | **1.26 ± 0.98** |
| Overall | | **13.26** | -5.14 | -2.74 | 9.01 | -1.05 | -18.20 | -15.71 |

*Table 12.* Backward Transfer ± std of different algorithms on task streams from LBF, CN, SMAC, SMACv2 and MAMuJoCo environments. The best and second-best methods are marked in **bold** and underlined, respectively. An asterisk (*) indicates that a method's mean performance is no lower than the best mean minus one standard deviation of the best method in the same row. All results are based on 5 distinct seeds and 32 episodes per seed on each evaluation step. "Overall" reports the average backward transfer of each algorithm over all task streams.

| Task Stream | Dataset | COMAD(ours) | FS | FT | EWC | OWL | Rehearsal | ODIS-FT | HiSSD-FT |
|---|---|---|---|---|---|---|---|---|---|
| LBF | Expert | -0.62 ± 0.77 | -98.13 ± 0.62 | -73.44 ± 4.06 | -89.69 ± 4.69 | -0.70 ± 0.63 | **0.46 ± 0.31** | -2.08 ± 1.35 | -0.75 ± 0.33 |
| | Medium | -5.62 ± 10.00 | -59.38 ± 6.88 | -49.38 ± 8.75 | -45.31 ± 7.81 | -0.78 ± 7.90 | **3.12 ± 3.12** | -3.50 ± 1.50 | 2.70 ± 1.30* |
| CN | Expert | **5.83 ± 0.83** | -10.83 ± 3.33 | -50.00 ± 0.00 | -8.75 ± 1.25 | -7.92 ± 6.25 | 1.25 ± 1.25 | -0.94 ± 1.56 | -3.80 ± 0.31 |
| | Medium | 1.25 ± 2.92 | **35.00 ± 4.17** | 33.75 ± 0.42* | 33.33 ± 2.50* | 0.83 ± 2.50 | 15.83 ± 10.00 | -7.19 ± 7.81 | 0.63 ± 1.88 |
| Marines | Expert | **30.80 ± 2.20** | 19.87 ± 6.03 | 21.21 ± 5.58 | 23.21 ± 12.50 | 30.21 ± 1.79* | -5.13 ± 29.24 | 0.31 ± 0.59 | 0.67 ± 0.46 |
| | Medium | **16.74 ± 8.71** | 8.33 ± 27.08* | -0.32 ± 6.70 | 2.68 ± 5.80 | 9.64 ± 11.72* | -14.84 ± 13.28 | 0.13 ± 3.66 | 2.90 ± 0.55 |
| Stalker-zealot | Expert | 16.67 ± 1.68 | -62.05 ± 34.82 | -23.44 ± 3.12 | -56.47 ± 40.40 | **21.65 ± 0.67** | 2.06 ± 3.40 | 0.27 ± 0.62 | 0.43 ± 0.12 |
| | Medium | 7.29 ± 6.42* | -7.37 ± 0.67 | -13.99 ± 0.60 | -6.92 ± 10.49 | **11.31 ± 9.23** | 7.81 ± 8.71* | -0.16 ± 0.12 | 0.13 ± 0.10 |
| Protoss | Medium | **8.59 ± 5.60** | -19.69 ± 1.56 | -16.25 ± 3.75 | -27.19 ± 3.44 | 2.19 ± 2.19 | -4.37 ± 4.37 | 2.81 ± 6.56 | 3.63 ± 4.78* |
| Zerg | Medium | **10.32 ± 4.49** | -12.40 ± 4.27 | -2.29 ± 6.46 | 4.17 ± 2.10 | 8.73 ± 5.60* | -16.09 ± 11.41 | 1.82 ± 1.30 | 1.74 ± 2.10 |
| Terran | Medium | **7.29 ± 1.93** | -2.81 ± 4.69 | 4.53 ± 8.56 | 4.69 ± 5.85 | 1.48 ± 12.31 | 4.79 ± 6.70 | 1.87 ± 1.63 | 1.82 ± 1.30 |
| Reward | Expert | **21.57 ± 11.20** | 4.19 ± 8.37 | 2.95 ± 4.55 | 10.53 ± 6.22* | 19.05 ± 5.43* | -0.70 ± 9.15 | 18.26 ± 7.07* | -2.86 ± 8.86 |
| | Medium | **16.13 ± 2.67** | 0.70 ± 2.75 | -8.72 ± 3.50 | -6.93 ± 0.71 | 10.67 ± 14.57 | -4.36 ± 3.29 | -4.08 ± 0.16 | -5.02 ± 0.81 |
| Dynamics | Expert | **11.32 ± 1.19** | -45.05 ± 4.30 | -50.93 ± 6.85 | -48.52 ± 6.93 | -0.20 ± 0.87 | -38.86 ± 2.18 | -51.09 ± 3.66 | -46.34 ± 4.23 |
| | Medium | 1.63 ± 3.17 | -38.16 ± 1.75 | -40.33 ± 4.98 | -42.94 ± 0.93 | **2.97 ± 1.30** | -34.19 ± 0.33 | -41.76 ± 5.36 | -31.46 ± 2.44 |
| Overall | | **9.95** | -19.19 | -17.78 | -16.94 | 7.28 | -5.55 | -5.69 | -5.04 |

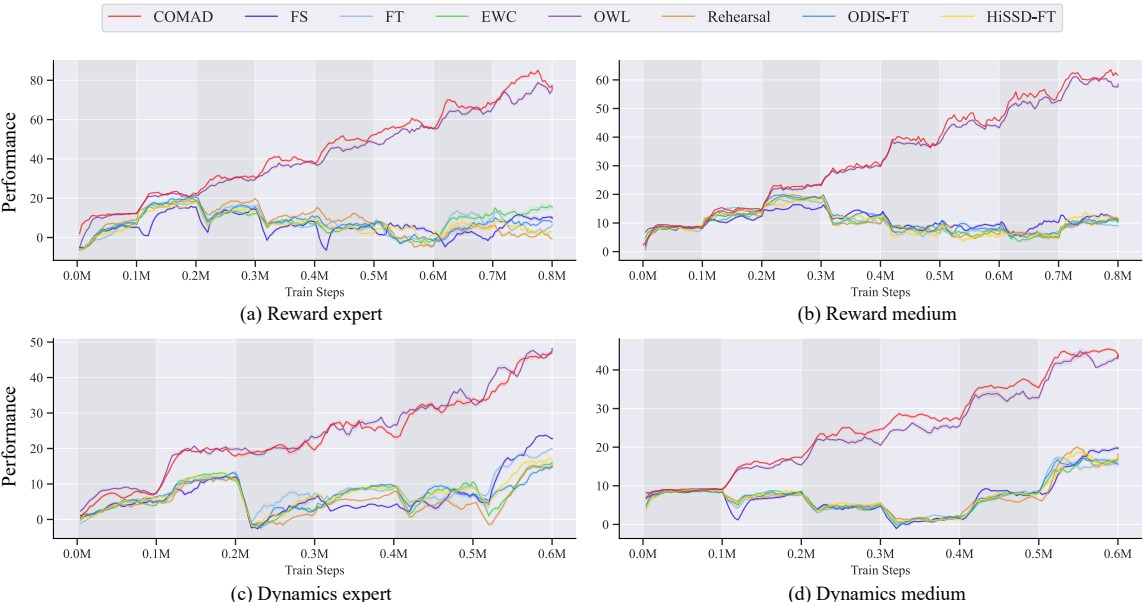

*Figure 12.* Learning curves on MAMuJoCo tasks.

