# OpenReview forum: "Offline Multi-agent Continual Cooperation via Skill Partition and Reuse"
_ICML.cc/2026/Conference — ICML 2026 regular_

### Official Review · Reviewer_QiYS · 2026-02-24

**Soundness:** 3
**Presentation:** 3
**Significance:** 3
**Originality:** 3
**Overall Recommendation:** 5
**Confidence:** 4

**Summary:**

The paper proposes COMAD, a framework for continual offline multi-agent cooperation via skill partition and reuse. The core idea is to learn coordination skills from offline datasets using a latent skill model, and then adapt these skills across a sequential stream of tasks. To handle distribution shift and avoid interference, COMAD employs a multi-head architecture that can expand when new skills are needed, along with a density-based estimator to decide whether previously learned skills can be reused. The policy objective is augmented to bias learning toward reusable skills while mitigating forgetting. Empirical results across several MARL benchmarks show improved performance, forward and backward transfer compared to prior offline and continual baselines.

**Compliance With Llm Reviewing Policy:**

Affirmed.

**Final Justification:**

I think (offline) continual MARL is quite an underexplored field, and deserves more attention. COMAD does a great job at addressing several challenges that plague agents when continually cooperating. Although the authors strongly build on existing work by integrating a continual skill discovery and reuse mechanism and a skill-augmented objective for transfer, their contribution of the reuse mechanism is noteworthy. COMAD is backed up by theoretical analysis and performs strong emprically. The authors extended their analysis from SMAC to MEAL during the rebuttal, further strengthening the evaluation. I had several comments about overclaiming/framing: (1) poor mention of limitations, (2) what part of the discussion is speculation vs. what is backed up by results, and (3) which components of COMAD are novel and which originate from prior works. The authors did not explicitly fix these issues in the rebuttal, likely due to the character limitation, but I am willing to believe this is not too difficult to revise in the paper. Similar to Reviewer bEpx, I was concerned about the growing heads issue. Not only because the tasks grow, but there don't seem to be any guardrails on the skill expansion mechanism. I am not fully convinced that the higher $d_0$ would limit the growth. However, from their experiment, it seems that practically this does not seem to be an issue, as the *n_head*s is much lower than I would have expected. Last, the authors replied that when including MEAL results, the average performance $P$ denotes the normalized number of delivered soups. The values of $P$ reported in their results table are up to 55. However, this is not possible, since that metric cannot exceed 2.0 with 2 agents. If the authors include these results in their camera-ready version, I think it is important to ask them to verify what metric they are reporting.

**Key Questions For Authors:**

1. How is the history length determined?
2. Since COMAD uses a multi-head architecture, are other CL baselines like EWC also allowed to use separate output heads?
3. Which part of the results interpretation in Section 5.3 is backed up by evidence, and which part is speculation?
4. How do we know the skill expansion doesn't go on uncontrollably, blowing up the size of the network?

**Limitations:**

There is no discussion of limitations whatsoever. The authors could be more open about the shortcomings of their work. Properly outlining limitations is important for setting readers' expectations and positioning the paper within the field.

**Strengths And Weaknesses:**

# Strengths

1. The architecture and objective are well-suited for continual learning. (1) The VAE information bottleneck works well for packing information and sharing representations between tasks. (2) The augmented advantage biases learning towards actions that old skills would have chosen. This is good for transfer. (3) The expandable multi-head outputs allow the skill space to grow. (4) The state-density estimator is a clever way to determine whether previously learned skills are applicable to the current task.
2. While the architecture is complex, consisting of many components, empirically, COMAD stands strong. The evaluation spans grid-based (LBF), 2D continuous (MPE, StarCraft), and 3D continuous-control (MA-Mujoco) domains. Combining skill discovery (SD) and regularization, COMAD beats CL methods without SD and SD methods not specialized for CL.
3. The key hyperparameters for COMAD appear not to be very sensitive. In particular, the choice of the NCE noise scale and the confidence threshold show robustness. Moreover, the authors use one set of parameters for all the environments.

# Weaknesses

1. A substantial portion of COMAD’s core appears to be inherited from prior offline multi-agent skill discovery works, ODIS [1] and HiSSD [2]. Section 4 introduces several key components as if they were novel contributions, instead of acknowledging connections to ODIS and HiSSD beyond a brief mention in the related work. (1) The global skill encoder from state-action pairs is already a central component of ODIS. (2) The overall autoencoder structure (latent skill representation + action decoder trained via offline RL objectives) closely mirrors the framework used in ODIS and, to some extent, HiSSD. (3) HiSSD also uses IQL-style objectives for offline policy/value learning. (4) The idea of separating reusable vs task-specific components also strongly echoes HiSSD’s common vs task-specific skill decomposition. Building on prior work is perfectly fine, but the authors should be more careful to delineate what is novel and what exists. That said, I think combining the “good parts” from ODIS and HiSSD, and adapting them for continual learning, is a strong contribution. It’s just that the paper currently portrays it as more novel than it actually is.
2. The evaluation regime is not clear. While Section 3 states that the authors adopt the *task-bounded* setting, it doesn’t explain whether the evaluated task is also known during inference. For instance, it has been shown that EWC’s performance heavily depends on having access to multiple output heads [3], which wouldn’t be possible in a task-agnostic setting. Although COMAD and OWL are allowed to leverage a multi-head architecture, it is unclear whether this extends to other baselines.
3. The skill expansion mechanism might indefinitely increase the number of output heads. A new head is added whenever the state-density overlap drops below a threshold, but it’s not clear what prevents this from happening all the time, perhaps if the data were very noisy. Even small distribution shifts could look “new” to the density estimator and trigger another head. Over time, this could lead to a steady growth in the number of heads, without any pruning or merging mechanism to keep things in check. The paper doesn’t really explore how sensitive this is to the threshold choice or how stable the density estimates are. Nor do the authors report the final number of heads per task stream or how often the expansion got triggered. So it’s hard to judge how well this would hold up outside the relatively controlled benchmarks with expert/medium datasets.
4. While the authors do a good job at extending several popular MARL environments to fit a continual learning setting, the evaluation would be stronger if COMAD were tested on a benchmark tailored particularly for the setting of Multi-agent Continual Cooperation [3], which is exactly what COMAD also targets.
5. The analysis in Section 5.3 is largely speculative and full of assertions. It is, of course, noteworthy that COMAD achieves stronger performance, but we don’t really get to see why that is. While Section 5.4 somewhat sheds some light on what is going on under the hood, the discussion in Section 5.3 is mostly speculation. It is difficult to tell whether COMAD performs better for the reasons claimed or whether other factors are driving the results. It is perfectly fine to have some degree of hypothezing when interpreting empirical findings, but the authors should more clearly outline which part of their interpretation is speculation and what claims they can back up with evidence.

### Minor Points

1. The middle section of Figure 1 is very difficult to read.
2. The caption of Figure 4 could better explain what the middle section of the figure with low, med, and high skill embeddings represents.

### Typos

1. 161 (left) individual policy —> individual policies
2. 161 (right) on global —> on the global
3. 182 (right) as scenario shifts —> as the scenario shifts
4. 194 (left) since agent —> since the agent / since agents
5. 194 (left) access to local —> access local
6. 196 (right) that utilize —> that utilizes
7. 215 (left) modeling sequence —> modeling sequences
8. 216 (left) single timestep —> single timesteps
9. 230 (left) facing new task —> facing a new task
10. 234 (left) are seen —> have been seen

[1] Zhang, Fuxiang, et al. "Discovering generalizable multi-agent coordination skills from multi-task offline data."

[2] Liu, Sicong, et al. "Learning generalizable skills from offline multi-task data for multi-agent cooperation."

[3] Tomilin, Tristan, et al. "MEAL: a benchmark for continual multi-agent reinforcement learning."

---

> ### Author Rebuttal · Authors · 2026-03-31
>
> Thank you for your constructive comments and suggestions. Below please find our response.
>
> **Q1: The authors should be more careful to delineate what is novel and what exists.**
>
> We thank the reviewer for this clarification. Our main contribution is to formulate and solve the problem of continual offline multi-agent skill reuse under task-bounded setting, by integrating a continual skill discovery and reuse mechanism and a skill-augmented objective for transfer. COMAD contributes a new continual learning formulation, a reuse mechanism and corresponding theoretical analysis compared to previous works.
>
> **Q2: The evaluation regime is not clear.**
>
> Our evaluation is under the task-bounded setting: the task ID is assumed to be known when training and evaluating a task, and is used to select the corresponding output head for multi-head methods. The task ID is not concatenated to the policy input but is only used for head routing and for deciding when to save parameter snapshots for regularizations (e.g., L2/EWC). In our current experiments, COMAD and OWL use multi-head architectures by design, while the other baselines are evaluated in their standard single-head forms as originally proposed in their respective literature.
>
> **Q3: The evaluation would be stronger if COMAD were tested on a tailored benchmark  (MEAL).**
>
> We appreciate this suggestion. We designed a task stream of 24 tasks within MEAL (200k transitions per task). As shown below, COMAD remains highly competitive on MEAL, achieving the highest FWT/BWT via its selective skill reuse mechanism.
>
> | MEAL | COMAD | MT | FS | FT | EWC | OWL | Rehearsal | ODIS | HiSSD |
> | --- | --- | --- | --- | --- | --- | --- | --- | --- | --- |
> | P | 53.58±4.68 | **55.33±2.63** | 16.32±5.46 | 10.48±3.71 | 49.32±7.64 | 51.23±5.31 | 50.25±4.87 | 27.61±7.86 | 24.32±5.22 |
> | FWT | **5.67±1.45** | N/A | N/A | -4.13±0.23 | 4.83±1.26 | 1.05±1.61 | 3.40±0.64 | 0.26±1.32 | -1.67±0.36 |
> | BWT | **1.40±2.26** | N/A | -25.37±4.24 | -30.62±6.87 | -1.52±4.47 | 0.36±2.54 | -2.74±1.51 | -23.07±8.92 | -15.84±6.59 |
>
> **Q4: Which part of the results interpretation in Section 5.3 is backed up by evidence, and which part is speculation?**
>
> We thank the reviewer for this rigorous distinction. We will revise Section 5.3 to explicitly separate evidenced facts (e.g., observed plasticity loss, memorization, COMAD/OWL's performance on tasks requiring explicit separation) from hypothesized underlying mechanisms (e.g., gradient conflict, feature oscillation).
>
> **Q5: How is the history length determined?**
>
> The history variable $\tau^i$ is the output of the Gated Recurrent Unit (GRU) and thus no history length needs to be specified.
>
> **Q6: Are other CL baselines like EWC also allowed to use separate output heads?**
>
> In our experiment setting, COMAD and OWL use multi-head architectures by design, while the other baselines including EWC are evaluated in their standard single-head forms. We ensure fair comparison by maintaining equivalent total parameter capacities where applicable.
>
> **Q7: How do we know the skill expansion doesn't go on uncontrollably?**
>
> To illustrate the growth of the skill library, we extend our experiment to repeated-CN-expert and Marines-expert and present the number of heads, the max reusability scores $\max_k \hat d_k$ and the reused heads as follows:
>
> | repeated-CN | CN-2 | CN-3 | CN-4 | CN-5 | CN-2 | CN-3 | CN-4 | CN-5 |
> | --- | --- | --- | --- | --- | --- | --- | --- | --- |
> | Average P | 24.82 | 48.13 | 64.76 | 78.32 | 76.47 | 77.41 | 79.71 | 78.95 |
> | n_head | 1 | 2 | 2 | 2 | 2 | 2 | 2 | 2 |
> | Max reuse score | N/A | 1.73 | 2.57 | 2.35 | 2.69 | 2.58 | 2.47 | 2.39 |
> | Reused head | new | new | CN-3 | CN-3 | CN-2 | CN-3 | CN-3 | CN-3 |
>
> | Marines | 3m | 4m | 10m | 12m | 5m_vs_6m | 7m_vs_8m | 9m_vs_10m | 13m_vs_15m |
> | --- | --- | --- | --- | --- | --- | --- | --- | --- |
> | Average P | 12.5 | 24.35 | 36.77 | 49.08 | 56.67 | 69.32 | 81.64 | 94.05 |
> | n_head | 1 | 1 | 2 | 2 | 3 | 3 | 4 | 5 |
> | Max reuse score | N/A | 8.56 | 5.83 | 9.34 | 4.62 | 8.07 | 6.47 | 5.53 |
> | Reused head | new | 3m | new | 10m | new | 5m_vs_6m | new | new |
>
> We observe that in CN($d_0=2$) only 2 heads are allocated and 3 reuses occur in Marines($d_0=8$), indicating that the number of heads grows controllably by setting the confidence threshold $d_0$. With higher $d_0$, the expansion will be more sparse and the growth can be effectively limited. We thank the reviewer for pointing this out and will append the sensitivity analysis of the threshold choice for head expansion in the revised version.
>
> **Q8: Minor Points and Typos:**
>
> We apologize for the mistake and poor readability, and will carefully revise the paper.
>
> **Q9: More discussion on limitations.**
>
> We thank the reviewer for pointing this out, and we will append more discussion on limitations in the revised version. Please refer to our response to **Reviewer N1EF (Q5)** for details.

---

> > ### Author Rebuttal · Reviewer_QiYS · 2026-04-03
> >
> > Thank you for your hard work. As a whole, I find the rebuttal very strong and satisfactory, and will increase my score. I have some final remarks/questions:
> >
> > 1. **Evaluation Regime (Q2)**. Thanks for the clarification. However, I don’t think it is a fair comparison if L2 & EWC are restricted to use multiple output heads. With a single head, they are solving a much more challenging problem. Namely, a single head removes the need to know how many tasks will be in the sequence, which is a likely scenario in practice. Just because in the original paper, they were presented without multiple heads, means they were also tackling this more complex task. Nevertheless, OWL is generally known to outperform EWC, with or without multiple heads. Therefore, since COMAD already outperforms OWL, I don’t consider this as a major issue.
> > 2. **MEAL Results (Q3)**. I think this addition makes the evaluation of COMAD stronger. I am curious about a few points. How did you collect the data on MEAL for offline training? Is it expert data, like on SMAC? What does the average performance P measure in your table? The returns? The delivered soup?

---

> > > ### Author Response · Authors · 2026-04-03
> > >
> > > We are glad that our rebuttal addressed most of your concerns. Regarding your final remarks/questions:
> > >
> > > 1. Thank you for this important nuance. We agree that single-head L2/EWC correspond to a more restricted regime, while COMAD/OWL are inherently multi-head methods in our task-bounded protocol. Our goal was to compare each baseline in its standard form rather than in an architecturally matched setting using the same head structure for all methods. We will clarify this more explicitly in the revised paper so that the scope of the comparison is stated precisely.
> > > 2. We collected the MEAL data using the official MAPPO implementation provided in the benchmark codebase. For each task, we trained MAPPO for 20M environment steps, by which point the policies had nearly converged, and then collected expert data from these policies. The average performance P in our table denotes the normalized number of delivered soups, following the metric used in the original MEAL paper.
> > >
> > > We again thank you for the thoughtful follow-up comments. They led to important clarifications of the evaluation protocol and the additional experiment, and will improve the precision and clarity of the revised paper.

---

### Official Review · Reviewer_N1EF · 2026-03-11

**Soundness:** 3
**Presentation:** 2
**Significance:** 3
**Originality:** 3
**Overall Recommendation:** 5
**Confidence:** 3

**Summary:**

This paper introduces COMAD, an approach for offline, continual multi-agent reinforcement learning of skills, such that skills learned on prior tasks can be reused while simultaneously learning new skills. In this cooperative setting, task-invariant coordination skills should emerge, leading to more performant agents. COMAD is described as two major components. In the first component, skills are discovered as skill embeddings from a global encoder, along with training a skill-conditioned policy / action decoder. Then, a local encoder is introduced to infer the skill embedding from local trajectory data. In the second component, a multi-head architecture is used to represent the skills learned in prior tasks. To reuse the skills learned in prior tasks, the advantage function is augmented with a reusability score that is calculated from the empirical state distribution via noise contrastive estimation. Experiments over a diverse set of 5 environments show that COMAD performs well overall. Theoretical insights for COMAD are also presented.

**Compliance With Llm Reviewing Policy:**

Affirmed.

**Final Justification:**

The two primary concerns I had with the paper, 1) how task ordering affects COMAD's performance and 2) the scaling of heads, have been addressed by the author response and new experiments. It is also good to hear that the authors will add more discussion of limitations and revise the paper writing. Therefore, I have raised my score, and my final recommendation is 5 (accept). The paper presents an important contribution to the offline MARL literature and has sufficient experimental rigor to justify the claims.

**Key Questions For Authors:**

1. What is the reasoning for choosing CN-expert and Marines-expert to conduct ablations in Section 5.5?
2. Is it possible to demonstrate how an online MARL method would succeed in these settings?
3. What is the effect of the ordering of the tasks? Do these have to be presented in a particular way? It seems as the tasks roughly correspond with difficulty, so what happens if the tasks are not ordered in a particular order?

**Limitations:**

There is some discussion of limitations, but the discussion is a bit light. It would also be helpful to expand upon these limitations in the paper.

**Strengths And Weaknesses:**

### Soundness
Overall, this paper shows good empirical evidence as shown in results across 5 diverse domains. The number of seeds is a bit low but is probably sufficient, although a greater number of seeds would further improve the rigor of the experiments. It is also helpful to see the hyperparameters in Table 7. However, I have two concerns with the version of the paper as written:
- I have a concern that sequencing of the tasks may play a factor in the results. By the framework of infinite stream of Dec-POMDPs, it does not appear to prescribe that the tasks are in any particular ordering, yet according to the tasks in the appendix, it does seem that the order of the tasks are generally (but not always) in difficulty or complexity order. It would help to better understand if COMAD's performance is sensitive to the ordering of the tasks. If so, then it would be helpful to revise the framework to note that the infinite stream must have a particular order.
- There is also another concern that the framework is the infinite Dec-POMDP stream setting, but the architecture uses a head per task. Therefore, I have concerns that the approach would not scale with the number of tasks, if the number of tasks becomes quite large that it becomes intractable.

### Presentation
The paper is generally well-written, although it is at times a bit difficult to parse.
- Figure 1 is useful to show the major components, but what is missing is a concrete description of the training pipeline. It may be helpful to have a new figure specifically that describes the algorithm.
- The first part of Section 4.3 may be better described at the beginning of Section 4. This would help with the narrative flow with then describing Section 4.1 and 4.2 as the major components of the pipeline, as currently it seems a bit out of place to have Section 4.3 where it is.
- The connection between Section 4.1 and 4.2 could be described more effectively in terms of how they go together, it feels a little disjointed.
- It may help to put a condensed version of Algorithm 1 in the main paper. It might be better to have this in the main paper and put the experiment in Section 5.2 in the appendix instead.
- Table 1: I would suggest underlining the second best approach as well and putting an asterisk if an approach is within 1 standard deviation of the best approach.

### Significance
The problem setting the authors are investigating is interesting and impactful. Learning from offline datasets would reduce costly online samples. The problem setting is also difficult: there are the typical multi-agent curse of dimensionality concerns, and there is the continual learning issues of plasticity and stability. So, this work appears to be useful for the field to further research in this area.

### Originality
The approach seems sensible to tackle the different distribution shifts that are occurring in this setting, and the empirical results suggest the approach is generally adept for the task (aside from any concerns regarding the number of tasks -> number of heads which have been mentioned in Soundness).

# Rebuttal
The author response in their rebuttal has addressed my two primary concerns. Therefore, I raise my "Soundness" score to 3 (good) and my review score to 5 (accept).

---

> ### Author Rebuttal · Authors · 2026-03-31
>
> Thank you very much for carefully reviewing our paper and providing constructive comments and suggestions. We are very glad that you appreciate our work. Below please find our response.
>
> **Q1: What is the reason for choosing CN-expert and Marines-expert to conduct ablations in Section 5.5?**
>
> The choice of CN-expert and Marines-expert in Section 5.5 is intended to analyze COMAD on two representative transfer challenges: the transfer under increasing agent population(CN), and the transfer in a more general and challenging cooperative scenario with heterogeneous task structure(Marines).
>
> **Q2: How an online MARL method would succeed in these settings?**
>
> Our problem focuses on the offline continual cooperation setting: the learner must handle distribution shift from fixed datasets, while also addressing interference, forgetting, and transfer across sequential tasks. Therefore, for a standard online MARL method to succeed here, it would need to resolve both the offline distribution shift issue and the continual adaptation issue. This is precisely the motivation for COMAD’s offline-constrained and expandable skill-reuse design.
>
> **Q3: What is the effect of the ordering of the tasks?**
>
> The task ordering usually has a significant effect on the performance of continual learning methods, which is confirmed in the literature[1,2], suggesting that an order of maximizing task dissimilarity is optimal. In our setting, proper task ordering may help learn more reusable skills. To investigate this, we conduct ablation studies on the task orderings of CN-expert and Marines-expert. Seq 0 stands for the original difficulty order, seq 1 reverses seq 0 (from harder to easier), seq 2 and seq 3 randomly scatter the task sequences:
>
> CN seq 2: "cn-3", "cn-5", "cn-2", "cn-4";
>
> CN seq 3: "cn-4", "cn-2", "cn-5", "cn-3";
>
> Marines seq 2: "12m", "3m", "7m_vs_8m", "13m_vs_15m", "4m", "9m_vs_10m", "10m", "5m_vs_6m";
>
> Marines seq 3: "5m_vs_6m", "10m", "13m_vs_15m", "4m", "9m_vs_10m", "3m", "12m", "7m_vs_8m";
>
> The results are listed as follows:
>
> | Marines-expert | Seq 0 | Seq 1 | Seq 2 | Seq 3 |
> | --- | --- | --- | --- | --- |
> | P | 94.05±0.83 | 95.66±4.96 | 93.36±10.68 | 96.75±4.61 |
> | FWT | 33.20±1.98 | 35.52±3.64 | 30.81±2.25 | 31.67±4.05 |
> | BWT | 30.80±2.20 | 27.52±5.93 | 13.43±2.68 | 24.72±5.16 |
>
> | CN-expert | Seq 0 | Seq 1 | Seq 2 | Seq 3 |
> | --- | --- | --- | --- | --- |
> | P | 78.49±0.41 | 84.27±0.49 | 82.81±2.81 | 88.13±1.25 |
> | FWT | 6.74±0.90 | 7.95±0.71 | 7.83±0.45 | 8.38±0.22 |
> | BWT | 5.83±0.83 | 6.27±0.31 | 6.05±2.92 | 5.96±3.75 |
>
> These results suggest that starting from medium-scale tasks (e.g., Seq 3 in both CN and SMAC) appears more beneficial than the difficulty order, indicating that these tasks may provide more reusable skills for later transfer. Nonetheless, COMAD remains effective for these four sequences, showcasing its robustness to the task ordering.
>
> >
> > [1] Li, Ziyan and Hiratani, Naoki. Optimal Task Order for Continual Learning of Multiple Tasks. ICML 2025.
>
> > [2] Bell, S. J., & Lawrence, N. D. The effect of task ordering in continual learning. *arXiv preprint arXiv:2205.13323*. 2022.
> >
>
> **Q4: The approach may not scale with the number of tasks if the number of tasks becomes quite large.**
>
> The frequency of head expansion in COMAD is controlled via the confidence threshold $d_0$: larger $d_0$ results in more reuse and more sparse expansion. We extend our experiment to investigate the growth of the skill library, please refer to our response to **Reviewer QiYS (Q7)** for details.
>
> **Q5: It would also be helpful to expand upon these limitations in the paper.**
>
> We further detailed our discussion on limitations and future work as follows and will expand them in the paper：
>
> First, the density-based reusability score acts as a proxy for the ideal skill-gating mechanism  given by the Lagrange multipliers(Appendix A.3). It captures whether current states fall within the support of previously learned skills. However, tasks in open-ended environments may require reusing skills based on fine-grained semantic features such as human feedback or LLM-based guidance, which is an important direction for future work.
>
> Second, the current work focuses on the task-bounded setting, allowing us to study transfer, forgetting, and skill expansion in a controlled way, but limits direct applicability to settings with unknown or ambiguous task boundaries. Extending COMAD to task-unbounded continual cooperation is therefore an important future direction.
>
> More broadly, evaluating whether the same skill partition-and-reuse principles continue to hold in more open-ended and semantically complex environments such as multi-agent embodied intelligence, remains an important next step.
>
> **Q6: The paper is generally well-written, although it is at times a bit difficult to parse.**
>
> Thank you very much for these constructive suggestions. We will add a clearer pipeline figure and re-organize the text as suggested in the revised version.

---

> > ### Author Rebuttal · Reviewer_N1EF · 2026-04-03
> >
> > Thank you for preparing the rebuttal response and for running these additional experiments.
> >
> > In my review, I noted two primary concerns: 1) how the task ordering affects the performance of COMAD, and 2) the scaling of heads. The sequencing experiment presented in this rebuttal addresses concern 1 (COMAD is robust to ordering), and the experiment presented for reviewer QiYS (Q7) addresses concerns 2. It is also good to hear that the authors plan to revise the pipeline figure and the text. Therefore, my issues have been fully resolved, and I will bring my score up to 5 (accept).

---

> > > ### Author Response · Authors · 2026-04-04
> > >
> > > Thank you very much for your careful review and encouraging feedback. We are glad to hear that all your concerns have been addressed.

---

### Official Review · Reviewer_bEpx · 2026-03-12

**Soundness:** 3
**Presentation:** 4
**Significance:** 3
**Originality:** 3
**Overall Recommendation:** 5
**Confidence:** 3

**Summary:**

This paper proposes the COMAD framework, which aims to address the challenge of "catastrophic forgetting" in multi-agent reinforcement learning when dealing with a continuous stream of different tasks. By utilizing a Variational Auto-Encoder (VAE), the framework extracts reusable coordination skills from offline data. It employs a multi-head architecture and a density-based reusability estimator to isolate knowledge from different tasks, ensuring that agents can effectively transfer and retain prior knowledge based on task similarity while simultaneously learning new skills.

**Compliance With Llm Reviewing Policy:**

Affirmed.

**Final Justification:**

The rebuttal has substantially improved my confidence in the paper’s technical contribution and positioning. I have raised my score.

**Key Questions For Authors:**

* Since the reusability score relies on state density, how does COMAD handle tasks that have different environmental appearances (states) but identical underlying coordination logic?
* As the task stream grows, the multi-head architecture leads to a linear increase in parameters. Is there a mechanism to merge or prune redundant heads that represent similar skills?
* Offline datasets often contain sub-optimal or "noisy" trajectories. How does the VAE ensure it extracts meaningful "coordination skills" rather than learning from failed experiences?
* This work is highly relevant to PSEC. Could you incorporate a discussion and comparative experiments with it in your paper?
> Tenglong Liu, Jianxiong Li, Yinan Zheng, Haoyi Niu, Yixing Lan, Xin Xu, and Xianyuan Zhan. Skill expansion and composition in parameter space. In The Thirteenth International Conference on Learning Representations, 2025.

**Limitations:**

* Repeated citation of the paper: Diversity is all you need: Learning skills without a reward function.
* The framework relies on state similarity to reuse skills. It may fail to transfer coordination logic between tasks that look different (different states) but require the same underlying strategy.
* Due to its multi-head architecture, the model's parameter count increases with every new task. This leads to "model bloat" and storage inefficiency in long-term, open-ended scenarios.

**Strengths And Weaknesses:**

Strengths
* Uses a "multi-head architecture" to isolate task-specific knowledge, preventing new data from overwriting old skills.

* Automatically calculates a "reusability score" to identify which past skills are relevant to new tasks, enabling effective forward transfer.

* The skill library expands dynamically as tasks increase, rather than being limited by a fixed capacity.

Weaknesses
* It relies on state similarity to reuse skills; if two tasks have different environments but identical logic, the system may fail to recognize the connection.

* As the number of tasks grows, the number of "heads" increases, leading to a higher total parameter count.

* Performance is strictly limited by the quality of the offline data; poor coordination in the dataset results in poor skill extraction.

---

> ### Author Rebuttal · Authors · 2026-03-31
>
> Thank you very much for carefully reviewing our paper and providing constructive suggestions. We are very glad that you appreciate our work. Below please find our response.
>
> **Q1: How does COMAD handle tasks that have different states but identical underlying coordination logic?**
>
> COMAD’s reusability score is introduced as a feasible proxy for the ideal skill-gating mechanism rather than a heuristic based only on raw state similarity (Appendix A.3), and is used in combination with the feature extractor. For tasks that differ at the raw-state level but share similar coordination logic, COMAD utilizes its population-invariant encoder to extract invariant coordination structure, reducing sensitivity to varying agent team and environmental appearances. If such invariances are captured, the resulting representations can still induce similar density estimates and trigger reuse. This is consistent with our empirical results on CN and SMAC: although these tasks differ in agent number and surface-level state statistics, COMAD still achieves strong forward transfer (Appendix D.3, Table 8), suggesting that reusable coordination patterns are being transferred beyond exact state matching.
>
> **Q2: Is there a mechanism to merge or prune redundant heads that represent similar skills?**
>
> COMAD utilizes the confidence threshold $d_0$ to control the growth of the skill library. By using higher $d_0$, head expansion at task arrival will be more sparse, limiting the size of the skill library. We extend our experiment to investigate the growth of the skill library, please refer to our response to **Reviewer QiYS (Q7)** for details.
>
> Currently we do not include an explicit merge/prune mechanism in COMAD, but it could be a promising direction for further adapting the output heads to selectively merge/prune based on the reusability scores and action distributions, after learning a few tasks.
>
> **Q3: How does the VAE ensure it extracts meaningful "coordination skills" from sub-optimal datasets?**
>
> In COMAD, the skills are derived from an advantage-weighted learning objective, where sub-optimal or failed actions receive smaller weights through the advantage weight term, rather than being copied uniformly from the dataset. On top of this, the latent skill bottleneck encourages the model to capture coordination structure that consistently explains the data, instead of memorizing individual noisy trajectories. Therefore, meaningful coordination skills arise from the combination of advantage-based reweighting and compressed latent skill representations.
>
> **Q4: Incorporate a discussion and comparative experiments with PSEC.**
>
> We thank the reviewer for suggesting a comparison with PSEC. Methodologically, the two works are related but have different emphases. COMAD focuses more on selective reuse across tasks in offline continual multi-agent cooperation settings, by identifying if the previously learned skills are compatible with the current task. This is exactly what our partition-reuse design and skill-augmented objective are intended to capture. Differently, PSEC focuses more on flexible skill expansion and composition in parameter space with LoRA modules, targeting diverse decision-making scenarios.
>
> At the same time, we agree with PSEC’s core insight that context-aware composition is important for leveraging prior knowledge, which is closely aligned with our motivation. The main difference is that PSEC assumes pretrained policies and fixed state/action spaces, which makes it less directly applicable to general multi-agent continual task streams, whereas COMAD is designed for offline continual multi-agent cooperation across evolving tasks.
>
> | Task Stream | Dataset | COMAD(ours) | MT | FS | FT | EWC | OWL | Rehearsal | PSEC |
> | --- | --- | --- | --- | --- | --- | --- | --- | --- | --- |
> | Reward | Expert | **77.75 ± 4.64** | 63.73 ± 5.93 | 10.18 ± 1.30 | 5.09 ± 1.06 | 15.45 ± 1.56 | 75.46 ± 5.08 | 5.33 ± 3.73 | 67.25 ± 2.50 |
> | Reward | Medium | **62.16 ± 3.22** | 30.24 ± 5.35 | 11.55 ± 2.06 | 9.01 ± 0.22 | 10.58 ± 1.00 | 58.15 ± 3.10 | 11.53 ± 2.13 | 61.96 ± 2.86 |
> | Dynamics | Expert | 46.66 ± 0.96 | **55.28 ± 4.27** | 23.24 ± 1.53 | 19.48 ± 0.53 | 15.65 ± 1.23 | 46.94 ± 4.31 | 14.77 ± 1.37 | 49.46 ± 2.09 |
> | Dynamics | Medium | 44.54 ± 3.73 | **53.14 ± 5.53** | 19.52 ± 1.05 | 15.72 ± 0.23 | 16.54 ± 0.63 | 42.87 ± 2.45 | 16.39 ± 3.78 | 39.04 ± 1.96 |
>
> MAMuJoCo is the only benchmark in our suite that is well aligned with PSEC’s setting, so we added the comparison there. The results show that PSEC is a competitive baseline and can transfer to offline continual multi-agent learning to some extent, but COMAD performs better overall: it is better on both Reward streams and on Dynamics-Medium, and is only slightly below PSEC on Dynamics-Expert.
>
> **Q5: Repeated citation of the paper: Diversity is all you need: Learning skills without a reward function.**
>
> Thank you for pointing out the mistake, and we will carefully revise the paper.

---

> > ### Author Rebuttal · Reviewer_bEpx · 2026-04-03
> >
> > Thank you for your response. I feel the rebuttal has substantially improved my confidence in the paper’s technical contribution and positioning. I am planning to raise my score.

---

> > > ### Author Response · Authors · 2026-04-03
> > >
> > > Thank you very much for your encouraging feedback and for recognizing the contribution and positioning of our work.

---

### Official Review · Reviewer_unJj · 2026-03-13

**Soundness:** 3
**Presentation:** 3
**Significance:** 3
**Originality:** 2
**Overall Recommendation:** 4
**Confidence:** 4

**Summary:**

This paper studies continual offline multi-agent reinforcement learning, where agents must learn cooperative policies from a sequential stream of offline task datasets while avoiding catastrophic forgetting and enabling knowledge transfer across tasks. The authors propose COMAD (Continual Offline Multi-agent Skill Discovery via Skill Partition and Reuse), a framework that discovers coordination skills from offline data using a variational autoencoder (VAE), partitions skills with a multi-head architecture to prevent interference between tasks, and selectively reuses previously learned skills through a density-based reusability estimator. The method augments the policy learning objective with guidance from reusable skills and theoretically connects this objective to the optimal solution of a constrained continual learning problem. Experiments on multiple multi-agent benchmarks, including Level-based Foraging, Cooperative Navigation, SMAC/SMACv2, and Multiagent MuJoCo, show that COMAD improves overall performance as well as forward and backward transfer compared to several offline skill learning and continual reinforcement learning baselines.

**Compliance With Llm Reviewing Policy:**

Affirmed.

**Final Justification:**

The authors feedback addresses my main concerns. I keep my positive score.

**Key Questions For Authors:**

1.Although the multi-head architecture helps mitigate interference between tasks, the number of parameters and computational cost will grow linearly with the number of tasks 𝑀. When the task sequence becomes very long (e.g., more than 50 tasks), is it still feasible to maintain such a large skill library? Have the authors considered mechanisms to compress, merge, or remove outdated or less useful heads?

2.The algorithm uses L2 regularization (Eq. 8) to constrain the feature extractor. However, the choice of λreg(set to 500) appears to have a significant influence on performance. In scenarios where tasks are highly heterogeneous, could such a strong constraint overly restrict the model’s plasticity when adapting to new tasks?

3.The paper compares with ODIS and HiSSD, but these methods were not originally designed for continual learning. If the authors could include comparisons with some recent continual learning methods, it would further highlight the advantages of the proposed skill partition design.

**Limitations:**

yes

**Strengths And Weaknesses:**

## Strengths

1.The paper not only proposes a novel algorithm but also provides theoretical analysis through Theorem 4.1, showing that the proposed Augmented Advantage Function can guide the policy toward approximating the theoretical optimum of the continual learning objective. The geometric mixture structure derived from the KKT conditions offers a rigorous interpretation of how skill reuse is achieved.

2.The framework adopts a multi-head (MH) architecture combined with a shared feature extractor, which effectively balances stability (preserving knowledge of previous tasks) and plasticity (learning new tasks).

3.The experiments cover a wide range of environments, from simple grid-world settings (LBF) to complex continuous control (Multiagent MuJoCo) and highly stochastic benchmarks (SMACv2). The method is compared against multiple baselines, including EWC, OWL, and recent MARL skill discovery approaches (ODIS, HiSSD), providing convincing empirical evidence of its effectiveness.

## Weaknesses
1.The proposed reusability score relies on state density estimation to determine whether previously learned skills should be reused in a new task. However, similarity in state distributions does not necessarily imply similarity in optimal coordination strategies or skills. Two tasks may share similar states but require substantially different policies due to different reward structures or objectives. The paper does not provide sufficient theoretical or empirical justification for why state density is an appropriate proxy for skill transferability.

2.The framework operates under a task-bounded setting where the boundaries between tasks are explicitly known. This assumption simplifies the continual learning problem, as the model can allocate or expand skill heads when a new task arrives. In more realistic continual learning scenarios, task boundaries are often unknown or ambiguous, and the ability of COMAD to handle such settings is unclear. This limits the general applicability of the proposed approach.

3.While the framework is well-designed, several key components are based on existing techniques, including VAE-based skill discovery, multi-head architectures for continual learning, and density estimation via noise contrastive estimation. The main contribution appears to be the integration of these components into a unified framework for continual offline multi-agent learning, rather than introducing fundamentally new algorithmic ideas.

---

> ### Author Rebuttal · Authors · 2026-03-31
>
> We thank the reviewer for the constructive feedback. Below are our responses.
>
> **Q1: Similarity in state distributions does not necessarily imply similarity in optimal coordination strategies or skills.**
>
> We clarify that our reusability score is a feasible proxy for the ideal skill-gating mechanism, rather than a naive heuristic based solely on raw state similarity. Theoretically, the ideal skill-gating is determined by the Lagrange multipliers in the optimal continual policy, weighting prior skills via both cross-task compatibility and state confidence (Appendix A.3). Directly computing these multipliers would require unreliable offline estimation of returns and KL terms due to extrapolation error.  Therefore, we introduce the state density-based score as a tractable approximation: familiar states provide a reliable support (confidence) for reusing prior skills. Crucially, the final skill selection is still advantage-driven through our augmented objective, not dictated by density alone. Section 5.4 empirically validates this design by showing the skill reuse process. Thank you for pointing out this ambiguity and we will revise the paper to make this theoretical motivation and empirical support more explicit.
>
> **Q2: The ability of COMAD to handle unbounded task settings.**
>
> We adopt the task-bounded setting to more clearly study skill transfer and forgetting across sequential tasks, and to develop the corresponding theoretical analysis in a unified framework. We also acknowledge that this limits the direct applicability of COMAD to settings where task boundaries are unknown or ambiguous, which is common in continual learning scenarios. Nonetheless, the density-based reusability estimator of COMAD suggests a natural extension to task-unbounded settings by monitoring whether incoming states fall outside the coverage of existing skills, and trigger reuse or expansion accordingly.
>
> Thank you for pointing this out. Unbounded continual cooperation is an important future direction and we will discuss it in the limitations and future work part.
>
> **Q3: The main contribution appears to be the integration rather than introducing fundamentally new algorithmic ideas.**
>
> Our contribution aims at the continual offline multi-agent setting: we formulate reusable skill learning under sequential tasks, integrate skill discovery with partition/reuse and a skill-augmented transfer objective, and provide theoretical analysis for this design. Therefore, we view COMAD as a principled continual-learning framework rather than a simple combination of existing components.
>
> **Q4:  The number of parameters and computational cost will grow linearly with the number of tasks.**
>
> We thank the reviewer for pointing out this core concern. Currently COMAD relies on the confidence threshold $d_0$ to control the growth of the skill library by reusing old heads with the highest reusability score above $d_0$. To maintain feasibility for very long task sequences, one can set higher $d_0$ for more sparse skill expansion. If necessary, COMAD can be configured to allocate only one head with a large enough $d_0$. We extend our experiment to illustrate this in our setting, please refer to our response to **Reviewer QiYS (Q7)** for details.
>
> Nonetheless, utilizing compression or merging mechanisms after learning some tasks is worth further investigation to enhance the per-task expansion evaluation. To achieve this, COMAD may provide a natural approach to evaluate the overlap of skills based on the reusability scores and action distributions, and to selectively compress/merge.
>
> **Q5: The choice of λreg(set to 500) could overly restrict the model’s plasticity.**
>
> $\lambda_{reg}=500$ indeed imposes a strong constraint on the shared feature extractor. However, in COMAD this constraint is applied only to the shared encoder, while task-specific adaptation is primarily handled by the multi-head policy architecture, whose parameters remain unconstrained. Intuitively, the regularization is used to preserve reusable shared structure, whereas plasticity for new tasks is mainly provided by the task-specific heads. Empirically, we do not observe a clear loss of plasticity even on heterogeneous task streams such as SMAC/SMACv2. In Appendix D.3 (Table 8), methods that constrain the whole model more directly (e.g., EWC) show substantially weaker forward transfer.
>
> **Q6: Compare with some recent continual learning methods.**
>
> We have added a comparison with PSEC on MaMuJoCo, highlighting COMAD’s advantages for coordination skill reuse in offline continual multi-agent cooperation scenarios. Please refer to our response to **Reviewer bEpx (Q4)** for details.
>
> > Tenglong Liu, Jianxiong Li, Yinan Zheng, Haoyi Niu, Yixing Lan, Xin Xu, and Xianyuan Zhan. Skill expansion and composition in parameter space. ICLR, 2025.
> >

---

> > ### Author Rebuttal · Reviewer_unJj · 2026-04-03
> >
> > Thank you for the rebuttal. It addresses my main concerns. I am happy to maintain my current score.

---

> > > ### Author Response · Authors · 2026-04-03
> > >
> > > Thank you very much for your careful review and feedback.

---

### Decision · Program_Chairs · 2026-04-30

**Decision:**

Accept (regular)

**Comment:**

The paper introduces a framework for continual offline multi-agent EL that discovers coordination skills via a VAE, partitions them across tasks using an expandable multi-head architecture, and selectively reuses previously learned skills through a density-based reusability estimator. The method is grounded in a theoretical connection between its augmented advantage objective and the conditions of the constrained continual learning problem. Experiments span multiple multi-agent environments, demonstrating strong overall performance and forward/backward transfer relative to both continual learning and offline skill discovery baselines.

On the other hand, several core components (VAE-based skill discovery, IQL objectives, multi-head continual learning) are inherited from prior work such as ODIS and HiSSD, and the paper would benefit from a more precise delineation of what is novel versus adapted. The task-bounded assumption and the absence of a head pruning or merging mechanism remain acknowledged limitations.

Overall though, the major concerns (task-order sensitivity, head scalability) were addressed, and new experiments on the MEAL benchmark were conducted, and so the remaining issues are primarily that of presentation. The paper makes a solid contribution to an underexplored area and should be accepted.